# The Hsp70-Hsp90 co-chaperone Hop/Stip1 shifts the proteostatic balance from folding towards degradation

Kaushik Bhattacharya[1], Lorenz Weidenauer[2], Tania Morán Luengo[3,4], Ellis C. Pieters[3,4], Pablo C. Echeverría [1,6], Lilia Bernasconi[1], Diana Wider[1], Yashar Sadian[5], Margreet B. Koopman[3,4], Matthieu Villemin[1], Christoph Bauer[5], Stefan G. D. Rüdiger [3,4], Manfredo Quadroni [2] & Didier Picard [1✉]

Hop/Stip1/Sti1 is thought to be essential as a co-chaperone to facilitate substrate transfer between the Hsp70 and Hsp90 molecular chaperones. Despite this proposed key function for protein folding and maturation, it is not essential in a number of eukaryotes and bacteria lack an ortholog. We set out to identify and to characterize its eukaryote-specific function. Human cell lines and the budding yeast with deletions of the Hop/Sti1 gene display reduced proteasome activity due to inefficient capping of the core particle with regulatory particles. Unexpectedly, knock-out cells are more proficient at preventing protein aggregation and at promoting protein refolding. Without the restraint by Hop, a more efficient folding activity of the prokaryote-like Hsp70-Hsp90 complex, which can also be demonstrated in vitro, compensates for the proteasomal defect and ensures the proteostatic equilibrium. Thus, cells may act on the level and/or activity of Hop to shift the proteostatic balance between folding and degradation.

[1] Département de Biologie Cellulaire, Université de Genève, Sciences III, 1211 Genève 4, Switzerland. [2] Protein Analysis Facility, Center for Integrative Genomics, Université de Lausanne, 1015 Lausanne, Switzerland. [3] Cellular Protein Chemistry, Bijvoet Center for Biomolecular Research, Utrecht University, 3584 CH, Utrecht, The Netherlands. [4] Science for Life, Utrecht University, 3584 CH, Utrecht, The Netherlands. [5] Bioimaging Center, Université de Genève, Sciences II, 1211 Genève 4, Switzerland. [6] Present address: European Association for the Study of the Liver, 1203 Genève, Switzerland. ✉email: didier.picard@unige.ch

Homeostasis of the proteome (proteostasis) is essential for health and longevity[1–3]. Proteotoxic stresses lead to protein misfolding and aggregation, which trigger protein quality control mechanisms. The major ones, all assisted by molecular chaperones, are protein refolding and the degradation of misfolded and aggregated proteins by the proteasome and by autophagy[2,4–6]. Defects in any of these mechanisms can cause severe proteotoxicity, which in turn can contribute to neurodegenerative disorders such as Huntington's, Parkinson's, and Alzheimer's, premature aging, and other diseases[1–3,5,7].

In eukaryotes, the Hsp70 and Hsp90 molecular chaperone machines are major contributors to proteostasis by providing a platform for folding of both nascent polypeptides and misfolded, structurally labile and mutated proteins, collectively called "clients"[8–14]. For folding and assembly of clients, both Hsp70 and Hsp90 undergo large conformational changes and collaborate with co-chaperones[11,15,16]. One of these co-chaperones is the Hsp70-Hsp90 organizing protein (Hop), encoded by the gene *STIP1* in mammals. It is an adaptor molecule between Hsp70 and Hsp90, which facilitates the folding, stabilization or assembly of clients by promoting their transfer to Hsp90 after the initial recognition and binding of clients by Hsp70 in collaboration with its J-domain containing co-chaperone Hsp40[16–18]. Hop forms a ternary complex with Hsp70 and Hsp90 using its tetratricopeptide repeat (TPR) domains. Two of its three TPRs, TPR1 and TPR2A, specifically bind the extreme C-terminal sequences EEVD and MEEVD of Hsp70 and Hsp90, respectively[18–20]. While these are the primary interaction surfaces, additional contacts serve to stabilize the complexes and to facilitate dynamic rearrangements[17,19,21,22].

Proteins, whose folding or refolding fails, are degraded by the proteasome, a highly conserved and regulated eukaryotic protease complex. It is a 1.6 to 2.5 MDa complex consisting of a 20S proteolytic core particle (CP) and a 19S regulatory particle (RP); the CP can be capped by one or two RPs resulting in 26S or 30S particles, respectively[23,24]. The RP is divided into a lid and a base and has unique regulatory functions; it recognizes ubiquitinated substrates produced by the E1-E2-E3 ubiquitination system, promotes their deubiquitination and unfolding, the subsequent gate-opening of the CP, and finally the loading of the processed substrates into the proteolytic chamber[25]. Dedicated chaperones for the assembly of CP and the RP base are well known, whereas RP lid assembly is still not well understood[24]. Hsp90 has been proposed to be an assembly chaperone for the RP lid complex based on genetic interactions in the budding yeast[26] and the reconstitution of the RP lid complex in *E. coli* co-expressing yeast Hsp90[27].

Prokaryotes and eukaryotic organelles do have Hsp70 and Hsp90 orthologs but lack a Hop-like protein; their Hsp70 and Hsp90 physically and functionally interact directly[28–31]. In eukaryotes, Hop is not absolutely indispensable as mutant budding yeast, worms (*Caenorhabditis elegans*), and flies (*Drosophila melanogaster*) are viable with only mild phenotypes[32–34]. In contrast, the deletion of *STIP1* is lethal early in embryonic development in the mouse[35], possibly indicating that the function of Hop might be cell type-specific or dependent on specific cellular states or requirements.

In this study, we have explored why Hop is present in eukaryotes, what its critical functions are, and whether and how the eukaryotic Hsp70-Hsp90 molecular chaperone machines may function without Hop to ensure proteostasis. Our studies on the functions of Hop as a co-chaperone of the Hsp70-Hsp90 molecular chaperone machines led us to the discovery of alternative cellular strategies that ensure proper protein folding and proteostasis in human and yeast cells lacking this co-chaperone. These findings highlight the persistence of evolutionarily more ancient mechanisms in eukaryotic cells that may contribute to balance protein folding and degradation under certain conditions.

## Results

**Human Hop knock-out cells maintain cellular fitness and proteostasis and are not hypersensitive to proteotoxic stress.** To study the functions of Hop in eukaryotic cells, we knocked out its gene *STIP1* in several human cell lines with the CRISPR/Cas9 technique. Quantitation of the *STIP1* mRNA of the knock-out (KO) clones by Q-PCR showed a drastic reduction (Supplementary Fig. 1a), and the absence of full-length Hop protein was confirmed by immunoblotting using a specific antibody to Hop (Fig. 1a). We did notice that the HEK293T clone KO1 expresses a residual low level of a truncated form of Hop, which we characterized by mass spectrometry (MS) (Supplementary Data 1); it only retains the Hsp90-binding domain TPR2A. In subsequent experiments, KO1 proved to behave essentially like the other HEK293T clone (KO33), which is devoid of any detectable trace of Hop. Morphological examination revealed no obvious differences between wild-type (WT) and KO cells (Supplementary Fig. 1b). Growth rates of KO cells are only moderately reduced (Supplementary Fig. 1c); this observation was supported by cell cycle analyses in that KO cells show less cyclic phase cells (S + G2/M) and more cells in the G0/G1 resting phase (Supplementary Fig. 1d). The flow cytometric analyses of cells stained with annexin V and propidium iodide (PI) showed that none of the KO cell lines have any obvious viability issues (Supplementary Fig. 1e).

A label-free MS quantitation of the whole-cell proteome indicated that the vast majority of the proteome is not affected by the KO (Fig. 1b and Supplementary Data 2). With our statistical cutoffs (see "Methods"), only a minor proportion of the identified proteins of HEK293T and HCT116 cells are altered by the KO. Gene ontology (GO) term analyses of the statistically significant changes reveal a strong enrichment of stress response processes and protein folding/refolding mechanisms in Hop KO cells (Fig. 1c). This prompted us to determine how the KO cells survive in various stress conditions. We exposed them to thapsigargin, DTT, and A23187, which all induce the unfolded protein response of the endoplasmic reticulum, and to the oxidative stress inducer $H_2O_2$. Remarkably, KO cells are clearly not hypersensitive to these stress agents since these lead to comparable numbers of dead cells (Fig. 1d) and to a rather reduced impact on cell morphology compared to WT cells (Supplementary Fig. 1f). Upon exposure to heat shock (HS), a stress which induces protein misfolding and aggregation, KO cells showed a similar sensitivity in the HEK293T background (Supplementary Fig. 1g), but a markedly reduced sensitivity in the HCT116 and A549 backgrounds (Fig. 1e). We also challenged both WT and KO cells with azetidine-2-carboxylic acid (AZC); the incorporation of this proline analog into nascent proteins causes misfolding and subsequent protein aggregation. KO cells are more resistant to this proteotoxic stress, based on the reduced number of apoptotic cells (Fig. 1f) and morphological examination (Supplementary Fig. 1h). Thus, the absence of Hop does not compromise the overall proteostatic equilibrium; instead, if anything, proteostatic buffering of KO cells becomes more robust and resilient to proteotoxic stress conditions.

**The proteasome and the Hsp70-Hsp90 molecular chaperone axis are differentially required in KO cells.** Cells cope with proteotoxic stresses by using two important protein quality control mechanisms: degradation of misfolded proteins by the proteasome and in some circumstances by autophagy, and refolding and stabilization of misfolded proteins by molecular

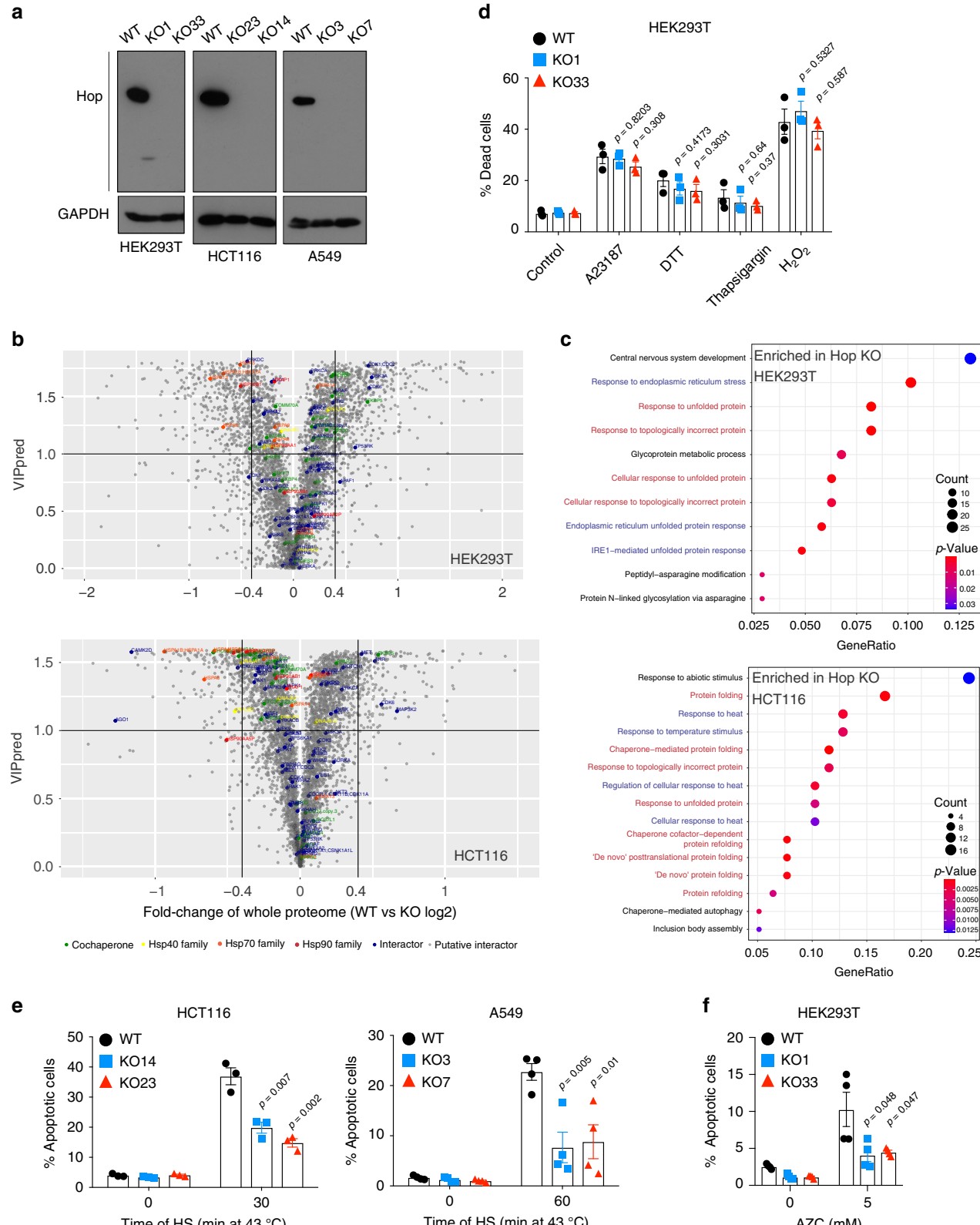

chaperones, including the Hsp70-Hsp90 molecular chaperone machines (Fig. 2a). To assess how these mechanisms are operating in KO cells, we pharmacologically blocked Hsp90, Hsp70, the proteasome, and ubiquitination. We observed that KO cells are hypersensitive to Hsp90 inhibition with geldanamycin (GA), as demonstrated by enhanced cell cycle arrest in the G2/M phase and accumulation of apoptotic cells (Fig. 2b and Supplementary

Fig. 2a). This observation was further confirmed by monitoring the effects of GA on the accumulation of dead cells (Supplementary Fig. 2b). KO cells also showed a higher sensitivity to the chemically different Hsp90 inhibitor PU-H71 (Supplementary Fig. 2c) and the C-terminal Hsp90 inhibitor novobiocin (Supplementary Fig. 2d). Remarkably, these findings are reminiscent of the observation with yeast that Hsp90 inhibition and Δsti1 are

**Fig. 1 Analyses of stress sensitivity and whole-cell proteome of human Hop KO cells. a** Immunoblot analysis of KO clones. **b** Volcano plot of the normalized fold changes of the whole-cell proteomes of WT and Hop KO cells. Proteins with VIPpred values >1.0 are variables of interest and cutoffs of >0.4 or < −0.4 are considered significant. VIPpred, the values of the variable importance in projection (VIP) predicted components (pred) were derived as mentioned in Methods. LFQ label-free quantification value. **c** GO term enrichment analyses of biological processes in Hop KO cells. Blue text, stress response-related terms; red text, protein folding/refolding-related terms. **d** Flow cytometric quantification of cell death induced by indicated reagents (*n* = 3 biologically independent samples). **e** Flow cytometric quantification of apoptotic cells after 48 h of recovery at 37 °C after HS for 0 to 60 min; for left and right panels, *n* = 3 and *n* = 4 biologically independent samples, respectively. **f** Flow cytometric quantification of apoptotic cells after treatment with AZC (*n* = 4 biologically independent samples). For the bar graphs, data are presented as mean values ± SEM. The statistical significance between the groups was analyzed by two-tail unpaired Student's *t*-tests. Source data are provided as a Source Data file.

synthetically lethal[36], indicating that this is an evolutionarily conserved characteristics. Similarly, the Hsp70 inhibitor JG-98 blocks the growth of KO cells more efficiently than that of WT cells (Fig. 2c and Supplementary Fig. 2e). We conclude that both Hsp90 and Hsp70 continue to be functionally required in KO cells, which appear to be even more dependent on them.

We checked the other side of the coin of proteostasis with proteasomal inhibitors. The treatments with both MG132 and bortezomib resulted in reduced cytotoxicity with all KO cell lines (Fig. 2d and Supplementary Fig. 2f, g). KO cells also displayed a higher resistance to the ubiquitin-activating enzyme inhibitor PYR41 (Fig. 2e and Supplementary Fig. 2h). These results suggested that KO cells are less dependent both on ubiquitination and proteasomal degradation. Moreover, transient overexpression of WT Hop in KO cells completely reversed the sensitivity to Hsp90 and proteasomal inhibitors (Fig. 2f–h). Since the proteostatic equilibrium could be influenced by changes in protein synthesis, we determined whether the latter is affected. A time-course experiment revealing puromycin-labeled and -released nascent chains showed that the translation rate is only minimally affected (Supplementary Fig. 2i). These results led us to hypothesize that proteostasis in KO cells is ensured by an alternative equilibrium between protein degradation by the proteasome and protein stability/refolding supported by the Hsp70-Hsp90. This raises two questions: (1) Is the function of the ubiquitin-proteasome system compromised in the absence of Hop? (2) How can the Hsp70-Hsp90 molecular chaperone machines function efficiently in the absence of Hop?

**The Hsp70-Hop-Hsp90 complex is physically and functionally associated with the proteasome**. To determine whether Hop must associate with Hsp70 and Hsp90 for optimal proteasomal function, we carried out immunoprecipitation (IP)-MS experiments with Hop mutants that are defective for Hsp70 or Hsp90 binding. Briefly, we used the TPR mutants K8A and K229A, which do not bind Hsp70 and Hsp90, respectively, and the double mutant[20], and confirmed the expected interaction patterns by IP (Supplementary Fig. 3a). Based on this result, we defined the Hop-specific interactome by an immunoprecipitation-mass spectrometry (IP-MS) analysis with transfected HEK293T KO cells. After an initial quality control by sodium dodecyl sulfate polyacrylamide gel electrophoresis (SDS-PAGE) (Supplementary Fig. 3b), samples were subjected to label-free liquid chromatography–mass spectrometry (LC/MS-MS) analysis. By comparison with the proteins associated with the TPR double mutant, ~41% were identified only with WT Hop and ~57% were enriched with WT Hop (Fig. 3a and Supplementary Data 3). Thus, proteins of the Hop interactome are largely dependent on the ability of Hop to bind Hsp70 and Hsp90 and may mostly be interactors of Hsp70 and/or Hsp90 rather than direct interactors of Hop itself. The components of the Hsp70-Hsp90 molecular chaperone machines are either enriched or only present with WT Hop, including Hsp70 (HSPA1), Hsc70 (HSPA8), Hsp90α (HSP90AA1), and Hsp90β (HSP90AB1)

(Fig. 3b). By comparison, Grp94 (HSP90B1), the endoplasmic reticulum-specific Hsp90 isoform, and not a known interactor of Hop, yields the lowest enrichment (~1.5-fold, Fig. 3b) in this subset of proteins. When we performed a GO term enrichment analysis, we observed that proteasomal and proteasome-associated ubiquitin-related proteins are overrepresented (Supplementary Fig. 3c–e), because these proteins are enriched or only present with WT Hop (Fig. 3c). Interestingly, out of the 19 identified proteasomal components, 16 are associated with the RP of the proteasome (Fig. 3c). We confirmed the interactions of several RP components with Hop-containing complexes by a targeted co-IP analysis with WT HEK293T cells (Fig. 3d). We reevaluated a published MS dataset of IPs of a panel of proteasomal components from yeast[37]. Remarkably, three of the bait proteins jointly pulled down Sti1 (yeast Hop), Hsp82/Hsc82 (yeast Hsp90s), and Ssa1/2 (yeast Hsp70s), compatible with the conclusion that the ternary complex associates with the proteasome in yeast (Supplementary Fig. 3f). Thus, the Hsp70-Hop-Hsp90 ternary complex is physically associated with proteasomal components, most notably with RP proteins.

The next experiments were designed to test whether the Hsp70-Hop-Hsp90 ternary complex is functionally relevant for the proteasome. We performed an in vitro proteasomal activity assay with extracts and noticed that the total steady-state activity of the 26S/30S proteasome is reduced across all KO cell lines (Fig. 4a and Supplementary Fig. 4a). In contrast, the rate of proteasomal activity is not significantly altered (Fig. 4b and Supplementary Fig. 4b). These results suggested that functional 26S/30S proteasome particles might be less abundant in KO rather than less functional on a per particle basis. We confirmed these in vitro results with an in vivo activity assay for the ubiquitin-proteasome system (UPS)[38]; this involved the transient expression and flow cytometric quantitation of a degradation-prone ubiquitin-GFP fusion protein (Ub-R-GFP) in parallel to its stable counterpart (Ub-M-GFP) as a control (Fig. 4c and Supplementary Fig. 4c). Upon transient expression of WT and TPR mutant Hop in KO cells, the in vitro proteasomal activity is rescued by WT Hop, whereas TPR single and double mutants are not as efficient and completely dead, respectively (Fig. 4d and Supplementary Fig. 4d). Interestingly, it had been shown that the genetic suppression of Hsp90 function in *Drosophila* shifts the protein degradation balance from proteasome-mediated degradation to autophagy[39]. However, this is unlikely to compensate for the proteasomal defect of Hop KO cells as the autophagic flux remains unchanged (Supplementary Fig. 4e); moreover, we did not see any upregulation of lysosomal or autophagy-related proteins in our whole-cell proteomic data (Supplementary Data 2). These results established that the Hsp70-Hop-Hsp90 ternary complex is not only physically associated with proteasomal components, but also functionally required for proteasomal activity.

Previous studies in different models had indicated that reduction or deletion of Hsp90 could negatively influence the proteasomal activity[26,39–41]. When we deleted the genes encoding

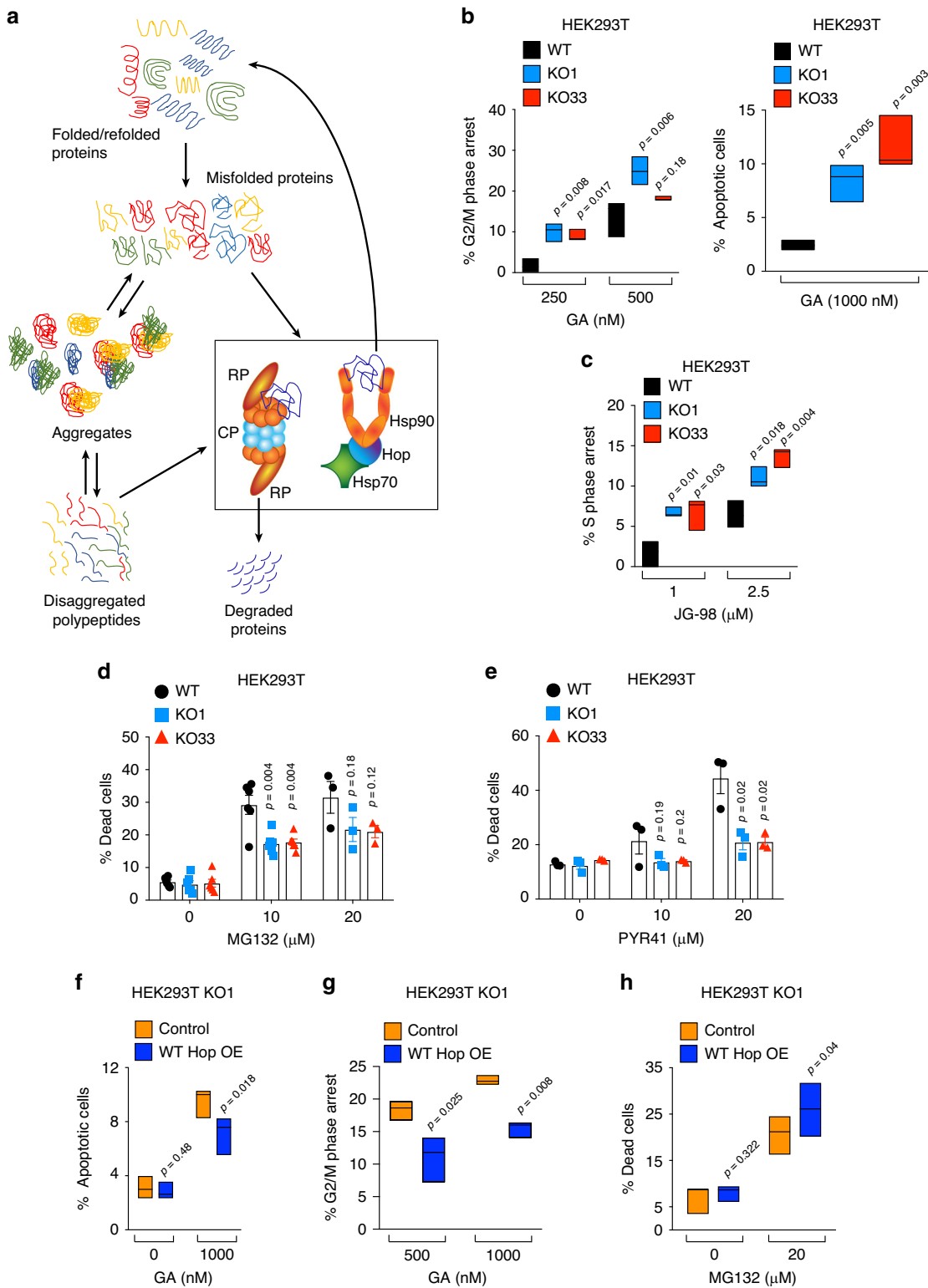

Hsp90α (*HSP90AA1*) or Hsp90β (*HSP90AB1*) in HEK293T cells, we found a reduced level of steady-state proteasomal activity (Supplementary Fig. 4f). The pharmacological inhibition of Hsp70 or Hsp90 in WT cells similarly led to a reduction of proteasomal activity, but suboptimal concentrations did not show any additive or synergistic effects (Supplementary Fig. 4g). However, proteasomal activity was not affected when GA was added to the extracts after cell lysis (Supplementary Fig. 4h). These results collectively indicate that functional Hsp70

and Hsp90, including the possibility to form a ternary Hsp70-Hop-Hsp90 complex, are essential for optimal proteasomal activity. Furthermore, the requirement for the ternary complex for full proteasomal activity is conserved in yeast: analogously to mammalian cells, the steady-state activity, but not the rate, is reduced in a Δ*sti1* strain (Fig. 4e and Supplementary Fig. 4i); note that the apparent slight differences in the rates of proteasomal activity are statistically not significant (Supplementary Fig. 4i).

**Fig. 2 KO cells are differentially dependent on proteasome and Hsp70-Hsp90 functions. a** Schematic representation of the proteostatic equilibrium relevant to this study. **b** Flow cytometric analysis of the GA-induced G2/M phase cell cycle arrest (left) and apoptosis (right). % G2/M phase arrest = % G2/M phase cells in inhibitor-treated sets—% G2/M phase cells in control experiment; % Apoptosis = % SubG0 phase cells in inhibitor-treated sets—% SubG0 phase cells in control experiment ($n = 3$ biologically independent samples). **c** Flow cytometric analysis of the JG-98-induced S-phase cell cycle arrest. % S-phase arrest = % S-phase cells in inhibitor-treated sets—% S-phase cells in control experiment ($n = 3$ biologically independent samples). **d**, **e** Flow cytometric analysis of the MG132- and PYR41-induced cell death. For panel **d**, $n = 3$–6 biologically independent samples depending on the treatment; for panel **e**, $n = 3$ biologically independent samples. **f**–**h** Flow cytometric analyses of GA-induced apoptosis and G2/M phase cell cycle arrest, and MG132-induced cell death of KO cells overexpressing (OE) WT Hop. Cells transfected with empty vector serve as a control ($n = 3$ biologically independent samples). For the bar graphs, the data are represented as mean values ± SEM. For the box plots, data are represented as the median values and edges of the box plots represent the range of the data. Statistical significance was analyzed by two-tail unpaired (panels **b**–**e**) and paired (panels **f**–**h**) Student's t-tests. Source data are provided as a Source Data file.

## Stability of individual proteasomal components is independent of the Hsp70-Hop-Hsp90 complex.

We considered two possibilities to explain the impact of the Hop KO on the proteasome: (1) Proteasomal components are clients of the ternary complex, and in the absence of Hop, they become unstable and subsequently degraded; (2) the ternary complex is required for 26S/30S proteasome assembly and/or maintenance. To address the first possibility, we reanalyzed published datasets of whole-cell proteomic experiments, where cells had been treated with Hsp90 inhibitors[42,43]. As expected, the levels of intracellular clients of Hsp90 are decreased and Hsp90-associated molecular chaperone and co-chaperone proteins are increased by Hsp90 inhibition (Fig. 4f). In contrast, Hsp90 inhibitors do not reduce the protein levels of any of the known core proteasomal proteins (Fig. 4f, right panel). We also experimentally checked the impact of GA on the levels of a few RP components in WT cells and found that none of them are strikingly diminished (Fig. 4g). Similarly, we did not find any striking and cell line-independent differences in protein levels of proteasomal proteins between WT and KO cells in our own whole-cell proteomic datasets (Fig. 4h). Thus, it seems unlikely that proteasomal components are dependent on Hsp90 for accumulation and/or stability.

## Overall composition of the assembled proteasome is similar in KO cells.

To evaluate the impact on proteasome assembly/maintenance, we purified 26S/30S proteasome particles. These were pulled out by an affinity purification strategy targeting the RP protein S5a (human gene name PSMD4); note that this scheme enriches for single- (26S) and double-capped (30S) CPs and discards free CPs or unassembled components. The integrity of purified 26S/30S was analyzed by native gel electrophoresis (Supplementary Fig. 5a, b), and by functional assays (Supplementary Fig. 5c, d). To characterize the composition of the purified proteasome, we performed a comparative label-free LC/MS-MS analysis. We did not find any consistent cell line-independent differences of any identified stoichiometric components of the proteasome between WT and KO cells (Supplementary Fig. 5e, f and Supplementary Data 4). Although we noticed that three substoichiometric components and proteasome chaperones were reduced in preparations from HEK293T KO cells (Supplementary Fig. 5f), we did not further consider them since they were not changed in HCT116 cells (Supplementary Fig. 5e). The aforementioned Hop IP-MS analysis showed that many proteasomal proteins are associated with WT Hop (Fig. 3c). Interestingly, in our own proteomic analyses of purified proteasome from WT cells, all components of the Hsp70-Hop-Hsp90 ternary complex could be identified, albeit at very low substoichiometric levels (Supplementary Data 4). Thus, the association of the Hsp70-Hop-Hsp90 ternary complex with the proteasome, while specific, may only be transient and regulatory. We concluded from these experiments that overall the composition of fully assembled particles is similar without Hop.

However, we cannot absolutely rule out the possibility that a minor fraction of purified free RP influenced the overall proteasomal composition.

## Optimal proteasomal assembly requires the Hsp70-Hop-Hsp90 complex.

We studied the structural integrity of the purified proteasome particles by negative staining transmission electron microscopy (TEM). Two-dimensional (2D) class averaging of all visible TEM structures led to four different proteasomal structural projections. We could see side views of single-capped 26S and double-capped 30S proteasome particles, and the two expected top views: an uncapped form with a central hole and a capped form corresponding to CP and RP face up, respectively (Fig. 5a, b). Based on the purification scheme and the relative levels of different forms in the native-PAGE analysis (Supplementary Fig. 5a, b), we did not expect any free CP. We measured the dimensions of all four proteasomal projections and confirmed that regardless of the presence of Hop, they are similar to the known values[44,45]. The main difference is the higher abundance of the double-capped proteasome (30S) in preparations from WT cells while the single-capped proteasome (26S) is more prevalent in preparations from KO cells (Fig. 5c); the opposite situation applies for the relative abundance of uncapped versus capped particles (Fig. 5d). Collectively, we concluded that the individual proteasomal components are not clients of the Hsp70-Hop-Hsp90 complex, but rather that it is involved in the capping/assembly and/or maintenance of the proteasome (Fig. 5e).

We next evaluated the ensemble of proteasome particles in whole-cell extracts by native-PAGE. We detected CP and 26S/30S proteasome particles using a specific antibody against the CP component Psma3, whose total levels are not significantly altered (Fig. 4h and Supplementary Fig. 6b). We found that the abundance of the 26S/30S particles but not CP is reduced in KO cells (Fig. 5f and Supplementary Fig. 6a, b). We further illustrated this finding by running the native-PAGE with 1.5-fold more total cell extract from KO cells next to an extract from WT cells (Supplementary Fig. 6c). As expected, only WT Hop and not the TPR double mutant can rescue proteasome assembly (Supplementary Fig. 6d, e). Note that the results of this rescue experiment and those with the purified proteasome (Supplementary Fig. 5a, b) argue against a technical problem underlying the low ratio of free CP to 26S/30S particles that we see with extracts from HEK293T and A549 cells (Fig. 5f and Supplementary Fig. 6a). Indeed, this ratio varies widely depending on the source[46–48], but independently of this, the difference is clearly genotype-dependent in our experiments. Moreover, the reduced proteasome activity in Δsti1 yeast is mirrored by the reduced assembly and/or stability of proteasome particles, but, as in mammalian cells, not the levels of the individual protein components (Fig. 5g and Supplementary Fig. 6f, g). The assembly defect of the proteasome in Hop KO cells is not due to altered subcellular localization of the proteasomal subunits either

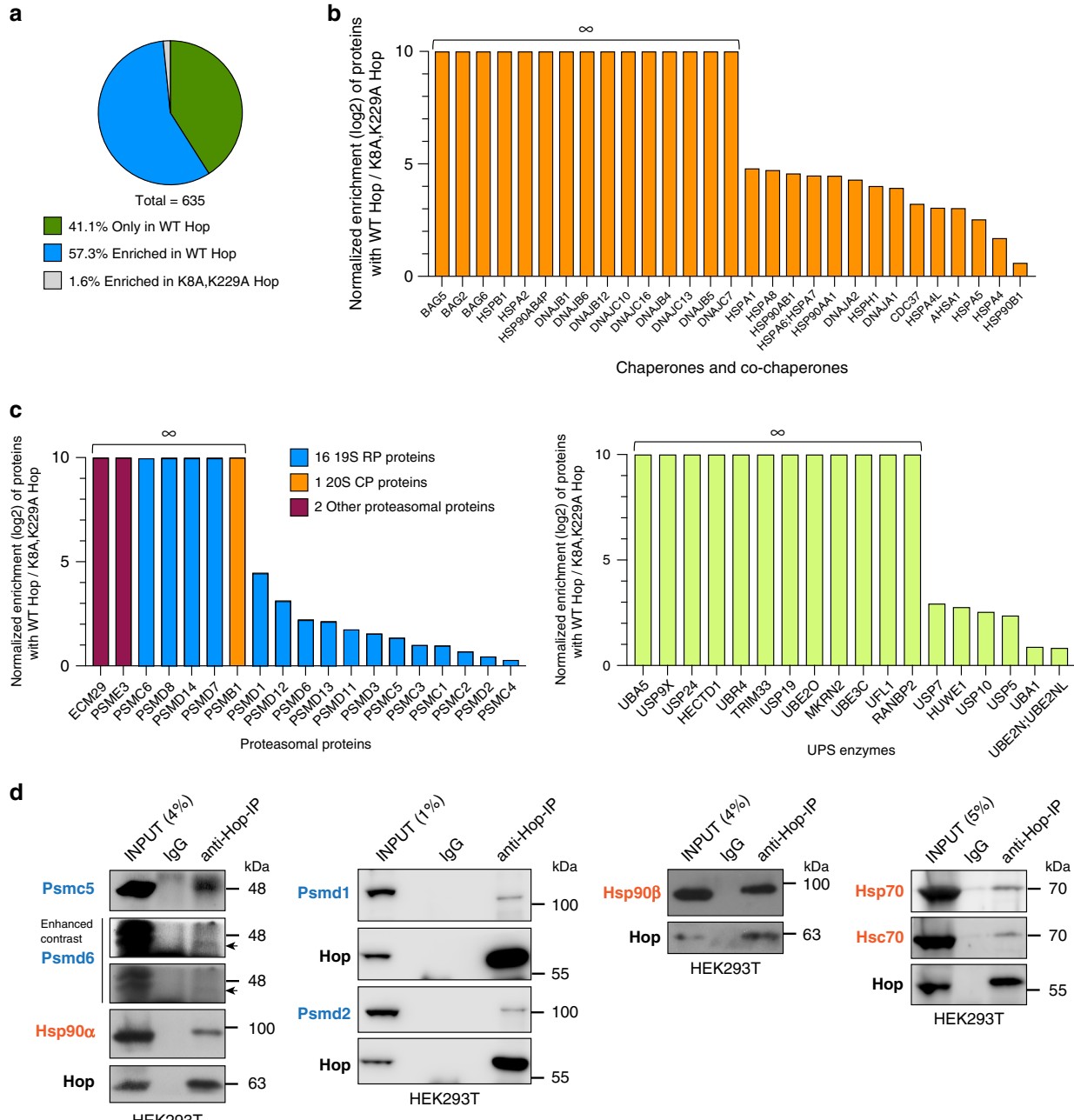

**Fig. 3 Hop is associated with proteasomal proteins. a** The pie chart represents the indicated groups of direct or indirect Hop interactors identified by IP-MS analysis of HA-tagged WT Hop and its TPR double mutant exogenously overexpressed in KO1 cells. **b** Bar graph with normalized log2 fold changes of Hsp70-Hsp90-related chaperone and co-chaperone proteins immunoprecipitating with the HA-tagged WT Hop compared to its TPR double mutant from KO1 cells by MS analysis as demonstrated in Supplementary Fig. 3b. ∞, only identified with WT Hop (*n* = 2 biologically independent samples). **c** Bar graphs as in panel **b** highlighting proteasomal proteins (left) and UPS-related enzymes (right). ∞, only identified with WT Hop (*n* = 2 biologically independent samples). **d** Validation of indicated proteasomal (in blue) and molecular chaperone (in orange) interactors of Hop in WT HEK293T cells by Hop-co-IP experiments. Normal IgG served as a negative control. For the Psmd6 immunoblot, the arrow indicates the band of the correct molecular weight. Validation of Hop interactors was performed with at least *n* = 2 biologically independent samples except for Psmd1 and Psmd2 (*n* = 1). For bar graphs, the data are represented as mean values. Source data are provided as a Source Data file and Supplementary data file.

(Supplementary Fig. 6h, i). Thus, the Hsp70-Hop-Hsp90 complex is essential for efficient proteasomal capping and/or for optimal proteasomal stability/maintenance (Fig. 5h).

**KO cells are less dependent on the proteasome even with proteotoxic stress.** So far, we had compared the Hop requirements for the proteasome under normal conditions. We now turned to investigate the effects of proteotoxic stresses. When cells

are treated with the Hsp90 inhibitor GA, more proteins form aggregates (Supplementary Fig. 7a), and during a HS, more insoluble and ubiquitinated material accumulates in KO cells (Supplementary Fig. 7b); this demonstrates that the reduction of proteasomal activity, and dependency of KO cells on it, is not a symptom of compromised ubiquitination. Upon inhibiting the proteasome in the recovery phase after a 6-h treatment with GA, we observed that WT cells are dying significantly more than KO

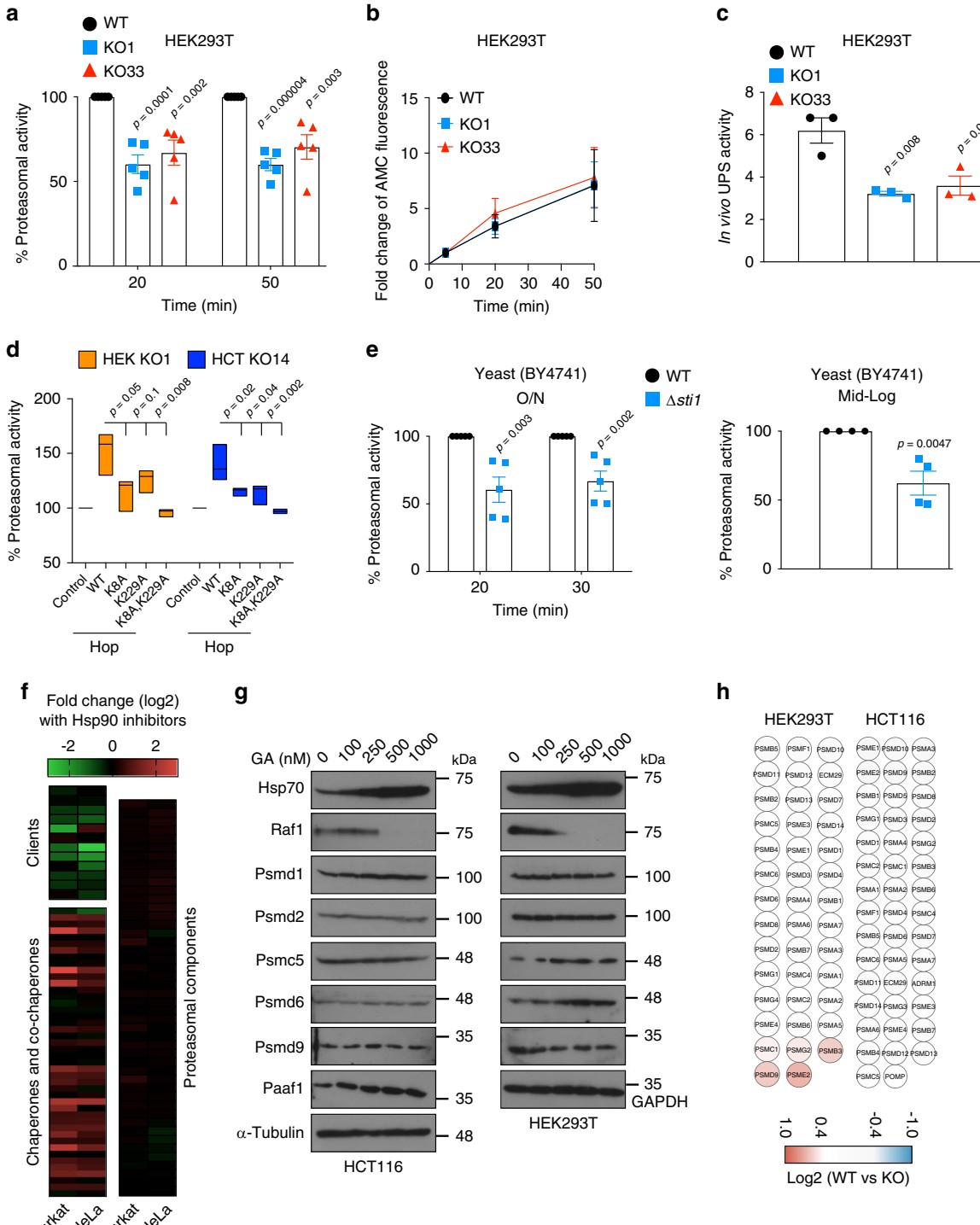

cells (Supplementary Fig. 7c-e). Similarly, inhibiting the proteasome after HS kills WT cells strikingly more than KO cells (Supplementary Fig. 7f, g).

We blocked the proteasomal activity with MG132 for 24 h and found that significantly less ubiquitinated substrates accumulated in KO cells (Supplementary Fig. 7h). This is consistent with a reduced degradation flux through the proteasome in KO cells since overall ubiquitination is not compromised and deubiquitination not augmented in KO cells (see above and Supplementary Fig. 7b). Intriguingly, an inverse and seemingly paradoxical relationship between the resistance to proteasome inhibitors, and proteasomal integrity and even flux had previously been

reported[47,49]. This further supports our conclusions that WT cells not only have a higher abundance of assembled proteasome particles, but that they are also more dependent on proteasomal function, and even more so in stressed conditions.

**Hsp70 and Hsp90 functionally collaborate without Hop in vivo.** Even though proteasomal assembly and function are compromised in KO cells, overall they nevertheless seem to maintain proteostasis (see Fig. 1b). To substantiate this conjecture, we biochemically fractionated soluble and insoluble proteins from WT and KO cells. Irrespective of genotype, we

**Fig. 4 The Hsp70-Hop-Hsp90 ternary complex is required for proteasomal activity. a** In vitro steady-state proteasomal activity with cellular extracts at the indicated time points after initiation of the reaction with the reporter substrate suc-LLVY-AMC. Activity of WT cells was set to 100% for each time point ($n = 5$ biologically independent samples). **b** Rate of proteasomal activity determined with suc-LLVY-AMC. The AMC fluorescence at 5 min was set as base value, and all other time points normalized to that ($n = $ at least 5 biologically independent samples). **c** Flow cytometric determination of in vivo UPS activity using the Ub-M-GFP and Ub-R-GFP reporter plasmids ($n = 3$ biologically independent samples). **d** In vitro steady-state proteasomal activity of extracts from KO cells overexpressing WT or TPR mutant Hop. The activity of mock transfected KO cells was set to 100%. HEK ($n = 3$ biologically independent samples) and HCT ($n = 4$ biologically independent samples), HEK293T and HCT116, respectively. **e** In vitro steady-state proteasomal activity of extracts of overnight (O/N, left panel; $n = 5$ biologically independent samples) and mid-log phase (right panel; $n = 4$ biologically independent samples) cultures of *STI1* (WT) and Δ*sti1* yeast cells (strain BY4741). Activity of WT cells was set to 100% for each time point. **f** Heat maps of normalized fold changes of the levels of Hsp90 clients and co-chaperones, and proteasomal components in Jurkat and HeLa cells treated with the Hsp90 inhibitors geldanamycin (20 h) and 17-DMAG (24 h), respectively. These MS data are from previous publications[42,43]. Overexpressed and downregulated proteins are in red and green, respectively. **g** Immunoblots of proteasomal subunits from GA-treated WT cells; Hsp70, Raf1, and α-tubulin serve as controls. **h** Heat maps of the normalized fold changes of the levels of proteasomal proteins identified by whole-cell MS analyses. The scale bar represents the log2 fold changes (WT vs KO) of the LFQ values. For the bar and line graphs, the data are represented as mean values ± SEM. For box plots, data are represented as the median values and edges of the box plots represent the range of the data. The statistical significance between the groups was analyzed by two-tail unpaired Student's *t*-tests. Source data are provided as a Source Data file.

found similar amounts of ubiquitinated proteins and protein aggregates (Supplementary Fig. 8a; see also untreated samples in Supplementary Fig. 7a, b). Therefore, more efficient chaperoning functions may compensate for the proteasomal defects of KO cells. We started to explore this with in vivo luciferase refolding assays; we found a higher rate of refolding of heat-inactivated luciferase in KO cells (Fig. 6a and Supplementary Fig. 8b), which could be reverted to that of WT cells by pharmacological inhibition of Hsp90 or Hsp70 (Fig. 6b and Supplementary Fig. 8c). Thus, Hop appears to restrain the refolding activity of Hsp70-Hsp90. Rescue experiments with overexpressed WT and TPR mutants of Hop in KO cells showed that only WT Hop inhibits luciferase refolding whereas the TPR mutants, and most significantly the double mutant, do not affect it (Fig. 6c). Hence, in the absence of Hop/Sti1, Hsp70 and Hsp90 are functional and responsible for enhanced folding of a model substrate.

From the literature, Δ*sti1* yeast cells are known to be heat-sensitive[32], unlike what we have found for human KO cells (Fig. 1e and Supplementary Fig. 1g). To check whether the Hop/Sti1-independent chaperoning mechanism is evolutionarily conserved in yeast, we performed an in vivo luciferase refolding experiment with the HS-sensitive Δ*sti1* yeast strain of the W303 background (Supplementary Fig. 8d). Despite its heat sensitivity, this strain displayed a faster rate of luciferase refolding during the early recovery phase (Supplementary Fig. 8e). To mirror the experiments with our HS-resistant human KO cells, we also performed the experiment with a Δ*sti1* strain of the BY4741 strain background, in which the mutant is as heat-resistant as the WT (Supplementary Fig. 8f). Here, the enhanced luciferase refolding in the absence of Hop/Sti1 is even more striking (Fig. 6d), and there is a much higher residual luciferase activity after a milder HS (Fig. 6e).

Another aspect of chaperoning is keeping aggregation-prone misfolded proteins in a soluble state. We checked the solubility of the aggregation-prone polyglutamine model protein Q74-EGFP[50] along with a non-aggregating control (Q23-EGFP). We observed a reduction of the aggregated form of Q74-EGFP in HEK293T KO cells (Fig. 6f and Supplementary Fig. 8g). This improved solubility seems to be Hsp90-dependent as it could be substantially suppressed by GA (Fig. 6g). In HCT116 cells, we did not see any difference between the two genotypes (Supplementary Fig. 8h), possibly indicating that cell lines may differ with respect to their intrinsic anti-aggregation activities. Intriguingly, these beneficial effects of the absence of Hop were recently identified by a genetic screen in *Drosophila*: a Hop knockdown was shown to reduce the aggregation of a Huntingtin mutant with a polyglutamine expansion[51].

Considering that cell viability and proliferation are relatively unperturbed and if indeed Hsp70 and Hsp90 function even better for some activities in the absence of Hop, we expected only a minimal impact on Hsp90 clients and co-chaperones. To test this hypothesis, we filtered our whole-cell MS dataset for proteins of the Hsp90 interactome (https://www.picard.ch/downloads/Hsp90interactors.pdf)[52]. The vast majority of the identifiable Hsp90 interactors are unaffected (Fig. 7a, b and Supplementary Fig. 9a, Supplementary Data 2). Interestingly, the heat-inducible isoform of Hsp70 (encoded by *HSPA1*) and Hsp110 (encoded by *HSPH1*), a nucleotide exchange factor (NEF) of Hsp70, are moderately upregulated in the absence of Hop (Fig. 7a and Supplementary Fig. 9a), whereas the levels of other co-chaperones of the Hsp70-Hsp90 chaperone systems, including several J-proteins, remain unaltered (Fig. 7a and Supplementary Fig. 9a).

To exclude the formal possibility that Hsp90 clients accumulate to normal levels but in an inactive conformation, we checked the activities of different classes of clients. As there are many tyrosine and serine/threonine kinases amongst the clients of Hsp90, we compared total levels of phosphorylated proteins; we could not see any significant global differences between the two genotypes (Supplementary Fig. 9b). We also did not observe any strong differences in the phosphorylation of specific sites indicative of activation of the Hsp90 client c-Src[53], and of its downstream kinases Erk1/2 (Supplementary Fig. 9c). Using luciferase reporter assays, we checked the transcriptional activities of several transcription factors, which are either Hsp90 clients by themselves or downstream of clients. We found that either the activity is enhanced (in HEK293T cells) or not strikingly compromised (in HCT116 cells) in the absence of Hop (Supplementary Fig. 9d). Thus, Hsp90 clients not only maintain their steady-state levels in KO cells, but they largely also maintain their activity compared to WT cells.

**Hop determines a unique spectrum of Hsp90 client proteins.** Since the vast majority of Hsp90 clients is unaltered in KO cells, we wondered whether challenging their proteostasis by over-expression of clients could reveal vulnerabilities. We over-expressed several steroid receptors, which need Hsp90 for proper folding, stability, and transcriptional activity[54–56]. Indeed, the accumulation and transcriptional activity of the glucocorticoid receptor is compromised in KO cells compared to WT cells (Fig. 7c, e, and Supplementary Fig. 9e). Accumulation and transcriptional activities of other steroid receptors, the estrogen receptor α and the progesterone receptor, are only moderately affected (Fig. 7c–e). v-Src, a viral tyrosine kinase client of Hsp90[57], accumulated to lower levels associated with reduced

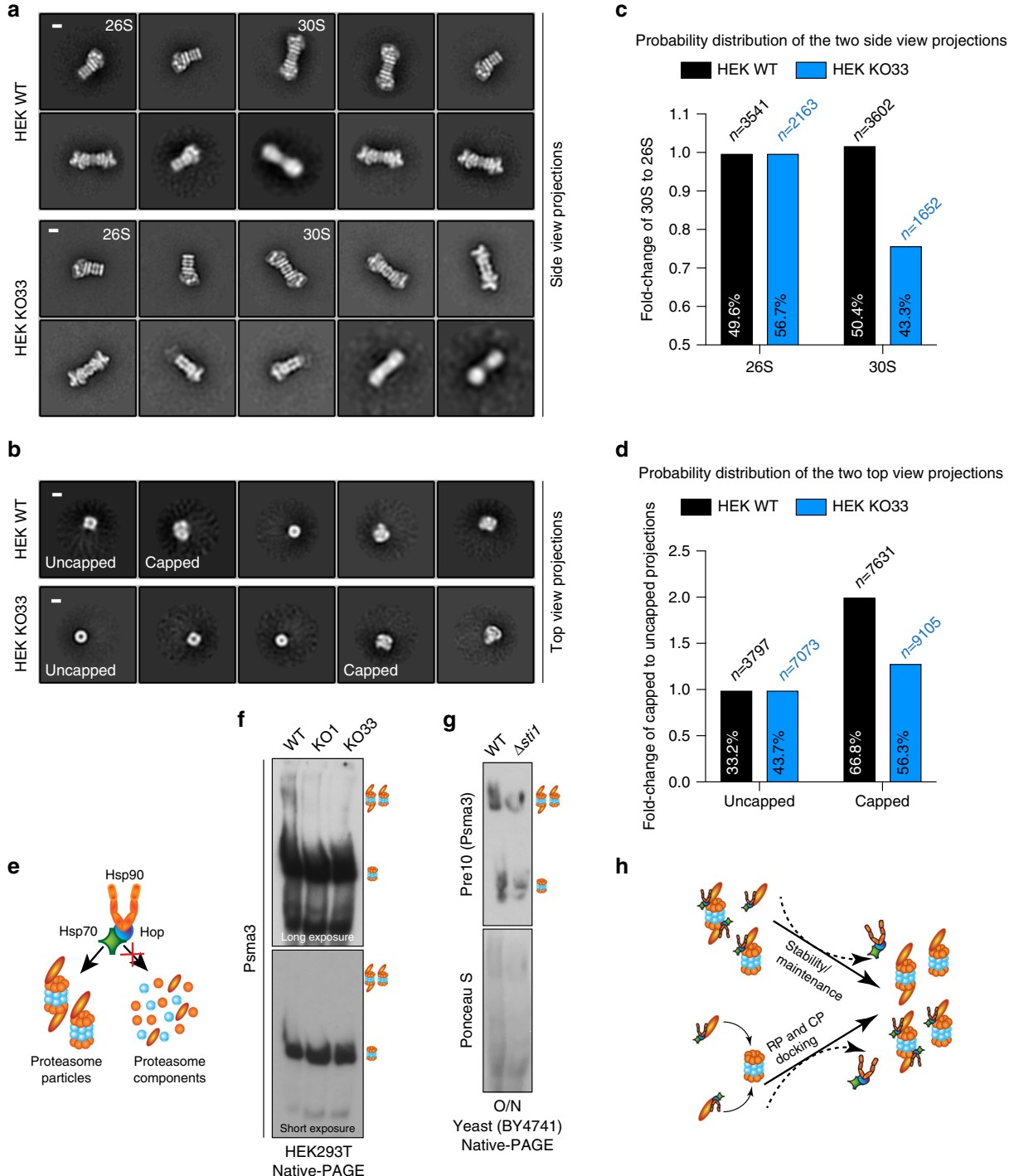

**Fig. 5 Hsp70-Hop-Hsp90 ternary complex is required for proteasome assembly. a, b** 2D class-averaged images of purified proteasome particles obtained by negative staining TEM. The top 10 and 5 classes are presented here for the side views (panel **a**) and top views (**b**), respectively, of both the WT and KO samples. Representative side views of double-capped 30S and single-capped 26S particles, and representative top views of uncapped (CP face up, central hole visible) and capped (RP face up, central hole invisible) particles are indicated. Scale bar, 10 nm. **c, d** Quantitation of the fold changes of 30S over 26S (related to panel **a**) and capped over uncapped (related to panel **b**) proteasome particles between in WT and Hop KO samples. n, total numbers of structural projections of proteasome particles statistically analyzed with the software Relion. **e** Schematic representation of the involvement of the Hsp70-Hop-Hsp90 complex in proteasome particle assembly/maintenance rather than for individual proteasomal proteins. **f, g** Abundance of different proteasomal particles of human (n = 5 independent samples over three independent experiments) and yeast cells (n = 5 independent samples over 2 independent experiments) as indicated displayed by 4% native-PAGE and subsequent immunoblotting (with antibodies against the indicated proteasome component). Positions of 26S/30S proteasome particles and free 20S CP are indicated using the bands of purified proteasome particles as standards. Different exposures, for a given cell line, are from the same immunoblot. Nitrocellulose filters stained with Ponceau S indicate equal loading of proteins. **h** Models of how the Hsp70-Hop-Hsp90 ternary complex could enhance the abundance and stability of assembled 26S/30S proteasome particles.

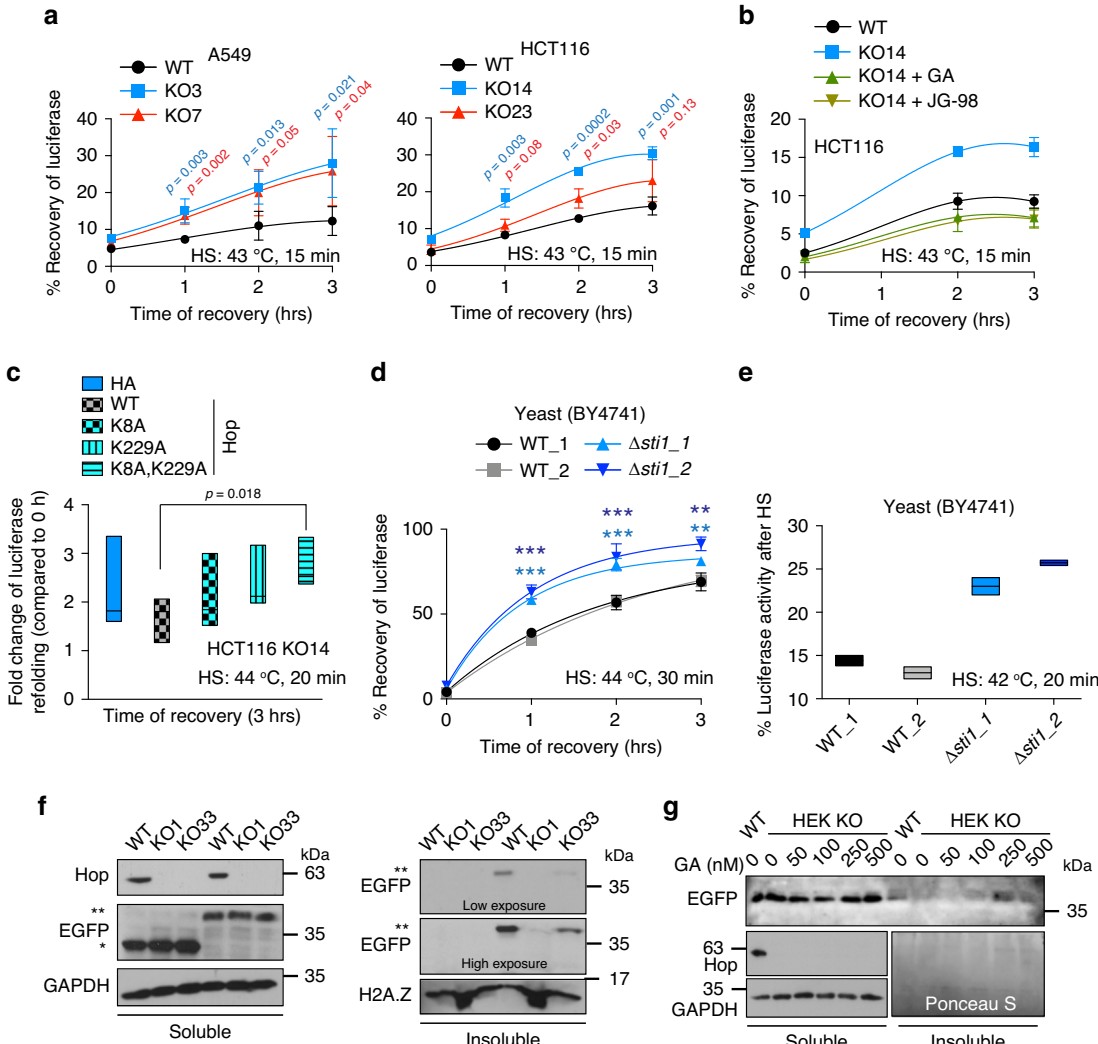

**Fig. 6 Hsp70 and Hsp90 are functional in vivo even without Hop. a** In vivo refolding of heat-denatured luciferase. Luciferase activity before HS is set to 100%. Left panel, $n = 4$; right panel, $n = 3$ biologically independent samples. **b** In vivo luciferase refolding in HCT116 KO cells treated with inhibitors before and during the recovery phase ($n = 3$ biologically independent samples). **c** In vivo luciferase refolding in HCT116 KO cells exogenously overexpressing WT or TPR mutants of Hop. Non-HS controls for each sample are set 1-fold ($n = 4$ biologically independent samples). **d** In vivo refolding of heat-denatured luciferase in WT and $\Delta sti1$ yeast cells (BY4741 strain background; two different transformants each). ***$p < 0.001$ and **$p < 0.01$ statistically significant differences between WT and $\Delta sti1$ ($n = 4$ biologically independent samples). **e** Residual in vivo luciferase activity immediately after mild HS ($n = 2$ biologically independent samples). Luciferase activity before HS is set to 100%. **f** Solubility of aggregation-prone polyglutamine model protein. Immunoblots of Q23-EGFP (bands marked with *) and Q74-EGFP (**) from soluble and insoluble protein fractions. GAPDH and H2A.Z serve as loading controls for soluble and insoluble protein fractions, respectively. **g** Immunoblot of Q74-EGFP from soluble and insoluble protein fractions of KO cells treated with GA. The Ponceau S staining of the nitrocellulose filter serves as loading control for the insoluble protein fractions. For the line graphs, the data are represented as mean values ± SEM. For box plots, data are represented as the median values and edges of the box plots represent the range of the data. The statistical significance between the groups was analyzed by two-tail unpaired Student's $t$-tests. Source data are provided as a Source Data file.

kinase activity in KO cells (Supplementary Fig. 9f, g). Several other Hsp90 clients (Hif-1α, Hif-2α, and androgen receptor (AR)) are not compromised upon overexpression in KO cells (Supplementary Fig. 9h). For endogenous GR as well, both expression and transcriptional activity are remarkably reduced in A549 KO cells (Fig. 7f). Overall, these results suggested that affected steroid receptors, kinases such as v-Src, and the proteasome are rather the exceptions amongst Hsp90 clients with regards to their pronounced Hop-dependence; anecdotally, GR and v-Src are also the very same Hsp90 clients that were the first ones to be discovered[54,57,58]. In agreement with a previous publication showing the plasticity of the Hsp90 co-chaperone network for folding of exogenous Hsp90 clients in yeast[59], we hypothesized

that Hop might determine unique criteria of client selectivity for Hsp90 even in higher eukaryotes.

**An alternative prokaryote-like Hsp70-Hsp90 binary complex maintains the Hsp90 interactome.** KO cells maintain proteostasis by more efficient Hsp70- and Hsp90-dependent chaperoning (Fig. 8a). Since bacterial and organellar Hsp90 and Hsp70 orthologs can interact without a Hop-like protein, we investigated the possibility that an alternative Hop-independent Hsp70-Hsp90 complex might form in the cytosol of human cells. With an Hsp90 IP-MS analysis, we found Hsp70/Hsc70, encoded by *HSPA1* and *HSPA8*, among the top hits even in KO cells, despite a 4-10-fold reduction (Fig. 8b and Supplementary Fig. 10a,

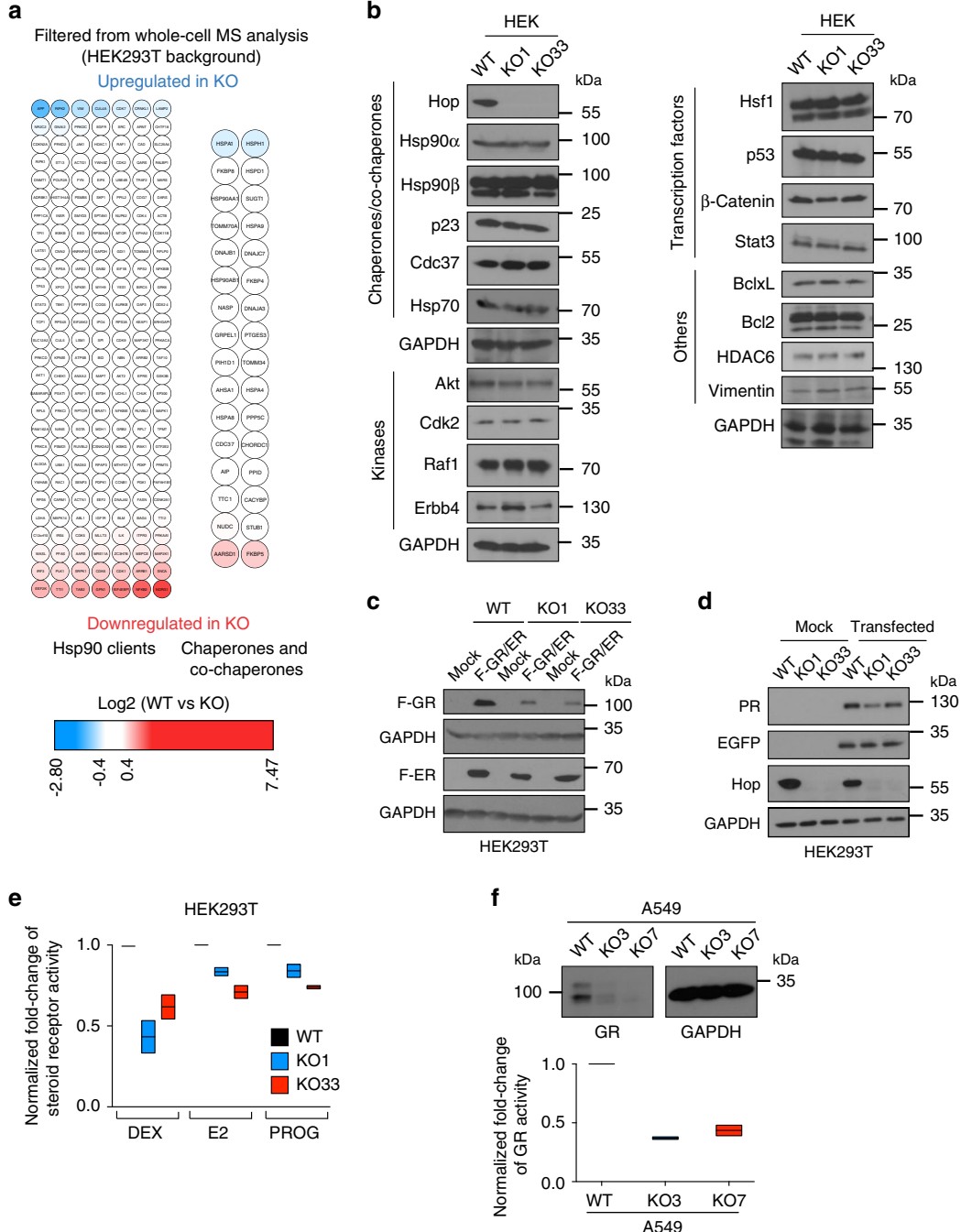

**Fig. 7 Hop KO only affects few specific Hsp90 clients. a** Heat maps of the normalized fold changes of the levels of Hsp90 clients (left), and molecular chaperones and co-chaperones (right) identified by whole-cell MS analyses. **b** Immunoblots of Hsp90 clients, Hsp90 partners and co-chaperones ($n = 3$ independent experiments). **c** Immunoblots of Flag-tagged GR (F-GR) and estrogen receptor α (F-ER) overexpressed in HEK293T cells ($n = 2$ independent experiments). **d** Immunoblots of the progesterone receptor (PR) overexpressed in HEK293T cells ($n = 2$ independent experiments). **e** Transcriptional activities of overexpressed GR (induced by dexamethasone (DEX)), ERα (induced by β-estradiol (E2)), and the PR (induced by progesterone (PROG)), as assayed with specific luciferase reporter genes ($n = 2$ biologically independent samples); data are represented as fold change relative to those of WT cells (set to 1). **f** Immunoblot of the endogenous levels of GR (top, $n = 2$ independent experiments), and its DEX-induced transcriptional activity determined with a transfected luciferase reporter (bottom, $n = 2$ biologically independent samples). For box plots, data are represented as the median values and edges of the box plots represent the range of the data. Source data are provided as a Source Data file.

Supplementary Data 2). We confirmed this Hop-independent interaction of Hsp70 and Hsp90 by a targeted co-IP experiment (Fig. 8c and Supplementary Fig. 10b).

In search of a mechanism promoting a Hop-independent interaction of Hsp70 and Hsp90, we checked whether Hop activity might be functionally redundant in human cells. We

revisited our own Hsp90 IP-MS datasets and extracted the data for proteins with a Hop-like architecture. There are only very few proteins with multiple TPRs, which are enriched with Hsp90 in the absence of Hop, and they are very moderately so (Supplementary Fig. 10c). Most importantly, all of these "enriched" TPR proteins are present at very low levels compared

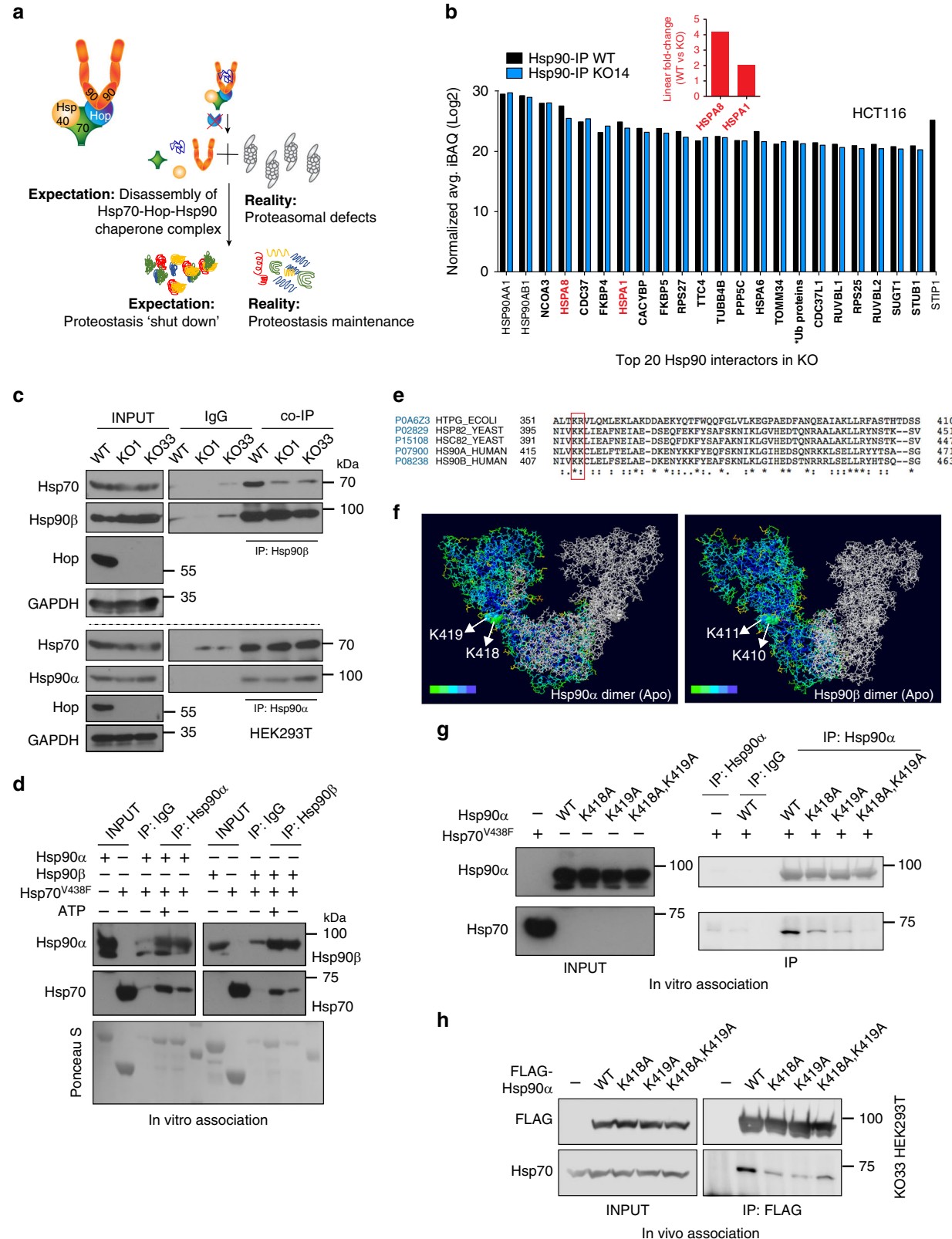

to the abundance of Hsp90 and Hsc70/Hsp70 in Hsp90 immunoprecipitates of either WT or KO cells, and to Hop in WT cells (Supplementary Fig. 10d). Thus, it is very unlikely that any of these proteins could substitute for Hop for the formation of another Hsp70-(multiple TPR protein)-Hsp90 ternary complex.

Using recombinant proteins, we tested the notion that human Hsp70 and Hsp90 could directly interact as they do in bacteria[28,29] and in Δsti1 yeast[60]. To minimize client-type interactions of Hsp70, we used the substrate-binding mutant V438F of Hsp70. The co-IP experiment revealed a direct interaction between Hsp90 and Hsp70 both in the presence and

**Fig. 8 Human Hsp70 and Hsp90 directly interact to form a prokaryote-like complex in the absence of Hop. a** Model of the impact of removing Hop on the Hsp70-Hsp90 molecular chaperone systems, the proteasome, and proteostasis. **b** Abundance of the top 20 Hsp90 interactors (highest iBAQ values). The graph shows the iBAQ values of the Hsp90 IP-MS analyses as log2. Hsp90α (HSP90AA1) and Hsp90β (HSP90AB1) were the bait proteins, and absence of Hop (STIP1) serves as quality control marker for KO cells ($n = 3$ biologically independent samples). Inset: Linear fold changes of the values for Hsp70 (HSPA1) and Hsc70 (HSPA8). *Ub proteins: UBB, UBC, UBA52, RPS27A. **c** In vivo interaction of Hsp90 and Hsp70 as determined by an IP experiment ($n = 3$ independent experiments). IgG, negative control IP with normal IgG. **d** IP experiment of in vitro interaction of purified recombinant Hsp90 and Hsp70 ($n = 2$ independent experiments). The substrate-binding mutant V438F of Hsp70 was used in this experiment. **e** Sequence alignments of yeast and human Hsp90 proteins with the bacterial Hsp90 HtpG. Evolutionarily conserved amino acids involved in the direct interaction of HtpG with Hsp70 (DnaK) in bacteria are highlighted by a red box. **f** Surface accessibility of the highlighted amino acids in the predicted dimeric human Hsp90 structures. The heat map represents a gradient of surface accessibility; most highly accessible amino acids are in green. **g** IP of in vitro interaction of purified recombinant WT or point mutant Hsp90α with Hsp70 (V438F) ($n = 2$ independent experiments). **h** IP of exogenously expressed FLAG-tagged WT or point mutant Hsp90α with endogenous Hsp70 ($n = 2$ independent experiments). For the bar graphs, the data are represented as mean values. Source data are provided as a Source Data file and a supplementary data file.

absence of ATP (Fig. 8d). We then checked the involvement of evolutionarily conserved surface residues that had been shown to be important for the direct interaction in bacteria[28] (Fig. 8e). As in bacteria, these residues are well surface-accessible in homology models of both human Hsp90 isoforms (Fig. 8f and Supplementary Fig. 10e), and modifying these residues in human Hsp90α strongly suppresses the interaction with Hsp70 in vitro (Fig. 8g) and with endogenous Hsp70 and Hsc70 in KO cells (Fig. 8h and Supplementary Fig. 10f).

Our whole-cell MS analysis had already indicated that the binary Hsp90-Hsp70 complex may be sufficient to support the Hsp90 interactome (Fig. 7a and Supplementary Fig. 9a). To address this issue more directly, we compared the levels of all proteins in the Hsp90 IP-MS datasets between WT and KO cells. The majority of the interactions do not change and there are similar proportions of proteins that are enriched or depleted in the absence of Hop (Figs. 8b, 9a and Supplementary Fig. 10a, g). Thus, client-Hsp90 and co-chaperone-Hsp90 interactions are largely maintained in KO cells, which supports the conclusion that the prokaryote-like Hsp70-Hsp90 binary complex in human cells is functional.

**The prokaryote-like human Hsp70-Hsp90 complex is also more efficient in vitro.** Substrate folding can be obtained by the collaboration of bacterial Hsp70 with Hsp90 in vitro[16,61]. To evaluate the chaperoning activity of the binary human Hsp70-Hsp90 complex, we performed in vitro luciferase refolding experiments with purified components. Human Hsp70 was complemented with a J protein, the NEF Apg2, and human Hsp90α. We measured luciferase refolding as a function of increasing concentrations of Hop. To our surprise, but in agreement with our in vivo experiments (Fig. 6a and Supplementary Fig. 8b), we discovered that the Hsp70-Hsp90 molecular chaperone systems refold heat-denatured luciferase most efficiently in the absence of Hop (Fig. 9b). Increasing concentrations of Hop gradually decrease the final yield achieved by the Hsp70-Hsp90 system (Fig. 9b). Moreover, only WT but not the Hsp70-binding mutant K418A,K419A of Hsp90α stimulates Hsp70-mediated refolding of luciferase in the absence of Hop (Fig. 9c), confirming the functionality of the human prokaryotic-like binary complex of Hsp70-Hsp90 (Fig. 9d).

## Discussion

The proteostatic equilibrium can be maintained by alternative mechanisms. For example, HS or inhibition of Hsp90 strongly induce the Hsp70-Hsp40 chaperone system and small Hsps to meet the new cellular requirements[62–64]. Here, we establish cellular models with human cell lines and yeast, which, in the absence of Hop/Sti1, adopt the more ancient and more efficient mechanism of chaperoning of bacteria and thereby compensate

for proteasomal defects, reestablishing an alternative proteostatic equilibrium. Thus, depending on cell-specific requirements or external inputs, eukaryotic cells may still be able to shift to a more prokaryote-like or organellar mode of operation for the Hsp70-Hsp90 systems. We speculate that the "invention" of Hop during the evolution from prokaryotes to eukaryotes may have promoted a shift from a proteostatic system centered on refolding to a more extensive use of proteasomal degradation.

Hsp90 itself had already been linked to proteasomal integrity and activity[26,27]. Our data complement this early evidence by demonstrating that it is the Hsp70-Hop-Hsp90 ternary complex that is required for optimal 26S/30S assembly and/or maintenance. According to current models, the RP ATPase-ring docks to the CP α-ring to form the functionally active holoenzyme[23,24]. The substoichiometric proteasome component Ecm29, which itself is not dependent on Hop (Supplementary Fig. 5e, f), has been claimed to be a tethering factor for RP and CP[65]. We propose that the ternary complex plays a similar role; further studies are required to define more clearly how the ternary complex promotes RP-CP docking and what the interplay with other factors might be. Our MS analysis showed that all components of the ternary complex are associated with 26S/30S particles at substoichiometric levels; hence, it is conceivable that the ternary complex acts like other *bona fide* proteasome-dedicated chaperones, which dissociate from the functionally mature proteasomal holoenzyme[66]. Although activity and assembly are known to be regulated by posttranslational modifications of its components[67], it is not likely that this can help to explain the proteasomal defects of KO cells; the vast majority of proteins, in general, and several of the known posttranslational regulators of proteasomal components such as MAPKs, mTORC1-related proteins, and protein kinase A, in particular, are unaltered in KO cells (Supplementary Data 2). Thus, we conclude that Hop, as part of the Hsp70-Hop-Hsp90 complex, primarily binds to the RP, facilitates the docking with the CP, and stabilizes the assembled 26S/30S proteasome.

Our most surprising finding is that cells lacking Hop/Sti1 compensate the proteasomal defect by improved protein folding. In view of the well-established role of Hop for substrate transfer between Hsp70 and Hsp90, and as allosteric regulator of the Hsp90 ATPase activity[17,22,68,69], this is the last thing one would have expected. Hop function may be beneficial or even essential to allow the folding or assembly of some substrates such as GR, v-Src, the proteasome, and possibly of some particularly labile clients such as the ΔF508 CFTR mutant[70]. TPR mutants of Hop, including the residual fragment of HEK293T KO1 cells, are unable to rescue Hop functions in KO cells. It should be acknowledged here that we do not have formal biochemical proof that Hop acts as part of a ternary complex in these contexts, but the fact that it must be able to form such a complex makes it very likely. In contrast to the aforementioned clients, for most Hsp90

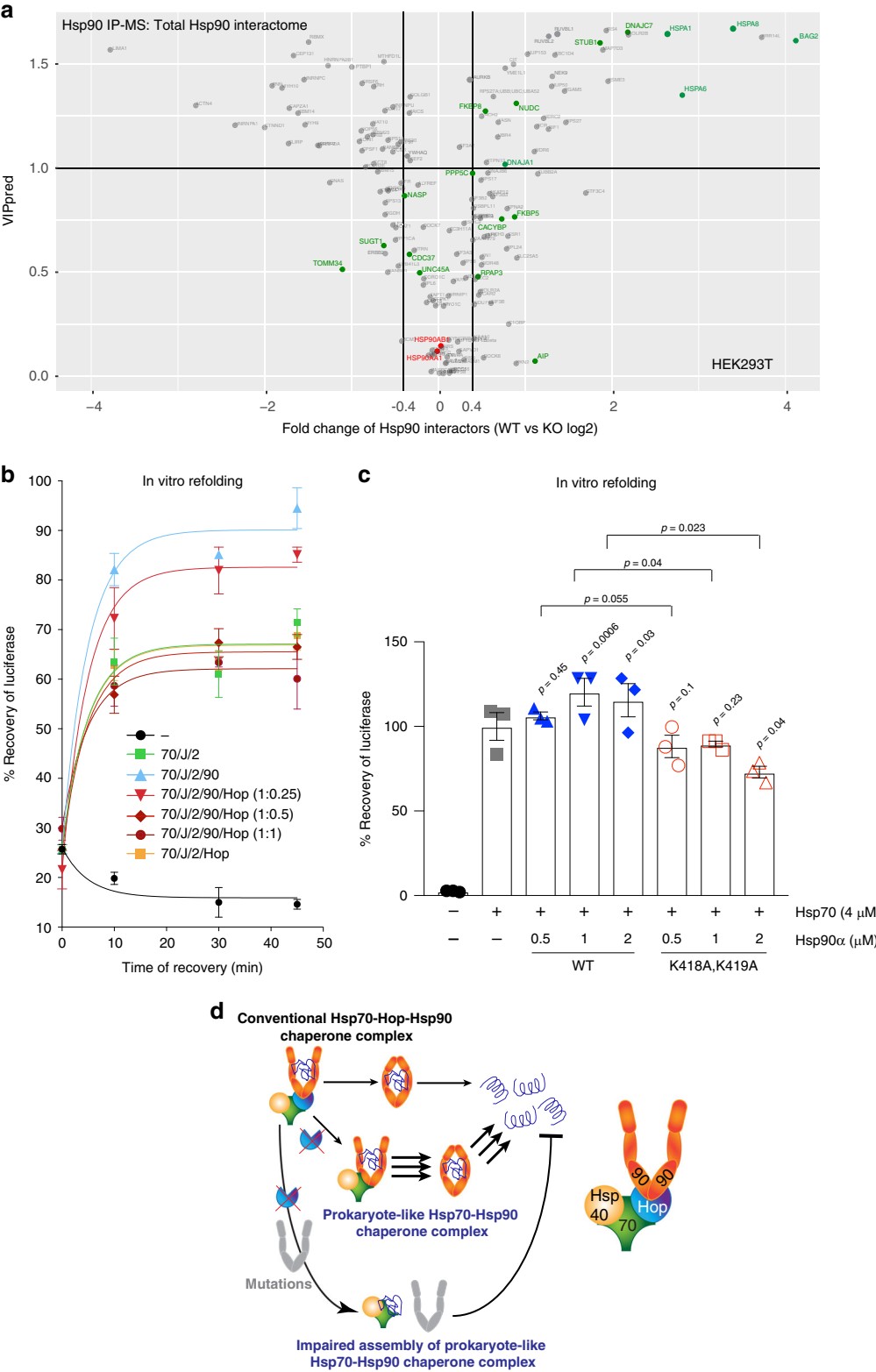

clients, the temporary slow-down of the Hsp90 molecular chaperone machine by Hop during substrate transfer may be counterproductive. What defines the Hop requirements of Hsp90 clients remains to be deciphered, but it is clear from our data that the vast majority of Hsp90 clients can be efficiently processed by the prokaryote-like Hsp70-Hsp90 binary complex. We assume

that this core molecular chaperone complex still requires the support of other co-chaperones like Hsp40, Cdc37, Aha1, and p23 to select, to process and to release Hsp90 clients[11,15,71–73]. Compromising the function of these co-chaperones could be particularly detrimental for Hop KO cells running on a somewhat hyperactive Hsp90 machine; indeed, the deletion of *SBA1*

**Fig. 9 Prokaryote-like Hsp70-Hsp90 complex is functional both in human cells and in vitro. a** Volcano plot of the normalized fold changes of the Hsp90 interactors identified by Hsp90 IP-MS. Proteins with VIPpred >1.0 are variables of interest and cutoffs of >0.4 or <−0.4 are considered significant. Hsp90 related co-chaperones are highlighted in green and the two cytosolic Hsp90 isoforms in red ($n = 3$ biologically independent samples). **b** In vitro refolding of heat-denatured luciferase by the indicated combinations of recombinant human Hsp90α (90), Hsp70 (70), DnaJB1 (J) and Apg2 (2) with different concentrations of Hop ($n = 3$ biologically independent samples). **c** In vitro luciferase refolding assay as in **b** comparing WT and mutant Hsp90α (K418A, K419A) in the absence of Hop ($n = 3$ biologically independent samples). The refolding yield in the absence of Hsp90 was set to 100%. This value was used as reference to calculate the indicated p-values that are not associated with a horizontal bracket. Statistical significance was determined by a two-tailed paired Student's t-test. **d** Model comparing the functions of Hsp70-Hsp90 complexes with and without Hop. For the bar and line graphs, the data are represented as mean values ± SEM. Source data are provided as a Source Data file.

(encoding the yeast p23 ortholog) in a Δ*sti1* yeast strain severely reduces cellular fitness[74].

Our data demonstrate that the Hsp70-Hsp90 binary system not only works, but somehow manages to be more efficient. Our quantitative Hsp90 IP-MS experiments show that the steady-state association of Hsp70 and Hsp90 is less prominent. We propose that the binary Hsp70–Hsp90 complex, albeit less stable, is more dynamic than the ternary complex. Moreover, similarly to what has been reported for the bacterial DnaK-HtpG system[28,75], the stimulation of the ATPase of one partner by the other might contribute to the improved chaperoning activity in the human system.

Whether Hsp70 and Hsp90 can form the binary complex in the presence of Hop in WT cells and whether this alternative molecular chaperone complex has any specialized functions in normal or stressed conditions remain open questions. Our data support this hypothesis since we observe a colocalization of Hsp70, Hsp90, Hsp40, and Hsp110, but not Hop, in HS-induced protein aggregates (Supplementary Fig. 10h). However, the specific detection of the Hsp70-Hsp90 binary complex and its characterization in the presence of Hop are technically extremely challenging. Further methodological developments will be necessary to explore its functions across different cell types and cellular conditions.

The highly proliferative state of embryonic stem cells is associated with rapid protein turnover and high proteasomal activity[76]. Deletion of some proteasomal subunits has been demonstrated to cause lethality in the mouse, flies and in plants[77–80]. It is, therefore, conceivable that the embryonic lethality of the Hop KO in the mouse[35] could be explained by the failure to meet this particular requirement in embryonic stem cells or in other rapidly proliferating cells during early development. The absence of Hop might also compromise the folding/ assembly of one or a few specific Hsp90 clients other than the proteasome, which are essential during early embryonic development. What the mouse model clearly demonstrates is that every eukaryotic cell may not be able to shift its proteostatic equilibrium to the overall more efficient chaperoning by the prokaryote-like binary complex in the absence of Hop. A systematic effort will be needed to determine whether Hop KO mouse embryos die because of a proteostasic collapse or because of a more subtle defect relating to some Hsp90 clients. Although we have established here that the absence of Hop does not affect proteostasis in cellular models, cellular or organismic fitness could nevertheless be negatively affected in the long run. Indeed, even though Hop KO worms are viable, their lifespan is reduced[33]. Furthermore, Hop levels drop in the aging brain, and aging is correlated with reduced proteasomal and chaperoning activity[67,81]. Since the absence of Hop generates a mixed outcome in different experimental models[32–35,82–84], tissue- and cell type-specific functions of Hop for the maintenance of proteostasis should be further studied.

Our discoveries may also have translational potential. If Hop could be specifically inhibited, this might promote superior chaperoning by the prokaryote-like Hsp70–Hsp90 binary complex.

This could be a useful strategy to reequilibrate the proteostatic balance in diseases or altered physiological states with proteasomal dysfunctions. Neurodegenerative disorders such as Huntington's and Parkinson's diseases, which may be associated with inefficient chaperoning[3,81] could potentially benefit from improved chaperoning induced by Hop inhibitors. Some compounds have been reported to inhibit specifically the interaction between Hsp90 and the TPR2A domain of Hop[85,86] and may be promising leads for such a therapeutic strategy.

## Methods

Additional methods are available in Supplementary Information.

**Reagents and resources.** Details on all reagents and resources are provided in Supplementary Data 5.

**Cell lines and cell culture.** HEK293T human embryonic kidney cells, HCT116 human colorectal carcinoma cells and A549 human lung epithelial carcinoma cells (as well as the corresponding Hop KO cell lines) were maintained in Dulbecco's Modified Eagle Media (DMEM) supplemented with GlutaMAX, 10% fetal bovine serum (FBS) and penicillin/streptomycin (100 μg/ml) with 5% $CO_2$ in a 37 °C humidified incubator.

**Yeast strains.** Yeast strain BY4741 (his3Δ1 leu2Δ0 met15Δ0 ura3Δ0) and its Δsti1 variant Y01803 (his3Δ1 leu2Δ0 met15Δ0 ura3Δ0 Δsti1::kanMX4), strain W303 (=BMA64-1A) (ade2-1 can1-100 his3-11,15 leu2-3, 112 trp1-1 ura3-1) and its Δsti1 variant W303 Δsti1 (ade2-1 can1-100 his3-11,15 leu2-3, 112 trp1-1 ura3-1 Δsti1::KanMX4) were maintained on yeast extract peptone dextrose (YPD) agar plates or in broth at 30 °C. To generate the W303 Δsti1 strain, W303 cells were transformed with a PCR fragment encompassing the Δsti1::KanMX4 locus of BY4741 and selected with geneticin (G418, 200 μg/ml) on YPD agar plates. Deletion of STI1 was verified by PCR amplification and by immunoblot analysis with a specific antibody to the Sti1-TPR2A/B domain. BY4741 and W303 yeast cells were transformed with pLG/LUC, a galactose-inducible luciferase expression plasmid, to produce yeast strains expressing firefly luciferase.

**Plasmids.** Site-directed mutagenesis was performed on pcDNA3.1(+)-Hop (WT) to generate human Hop mutants K8A (codon change: AAG > GCG), K229A (AAA > GCA), and the double mutant K8A/K229A, each with a C-terminal HA-tag[20]. Site-directed mutagenesis was performed on the bacterial expression vector pCA528-Hsp90α (WT) and mammalian expression vector pFLAG-CMV2/Hsp90α (WT) to generate the plasmids for the human Hsp90α mutants K418A (codon change: AAA > GCA), K419A (AAA > GCA), and the double mutant K418A/ K419A, each with either a 6x-His or FLAG tags. Sequences of all oligonucleotides (from Microsynth) are described in Supplementary Data 5.

**Genome engineering.** Hop (STIP1) KO HEK293T, HCT116, and A549 cells, as well as Hsp90α (HSP90AA1) and Hsp90β (HSP90AB1) KO HEK293T cells were generated by the CRISPR/Cas9 gene editing technology. The guide RNAs (gRNAs) were identified by and the corresponding oligos designed using the ATUM (previously DNA2.0) CRISPR/Cas9 design tool (https://www.atum.bio/). The gRNA sequences corresponding to the 5th exon of STIP1, 1st exon of HSP90AA1, and 6th exon of HSP90AB1 are listed in Supplementary Data 5. Sense and antisense oligos were synthesized (Microsynth), annealed and cloned into the BbsI site of px459 (Addgene plasmid #48139) as described previously[87]. WT HEK293T, HCT116, and A549 cells were transiently transfected with the gRNA plasmids using polyethylenimine (PEI). At 48 h post-transfection, transfected cells were selected with puromycin (3–5 μg/ml). Surviving cells were further cultured in the absence of puromycin until visible cellular foci formed. Cellular foci were individually picked and analyzed by immunoblotting using primary antibodies specific to Hop, Hsp90α, and Hsp90β. Clones which did not express Hop, Hsp90α or Hsp90β were

considered KO cells and frozen in liquid nitrogen. Some KO clones were also validated by MS analysis. The frequency of obtaining Hop KO clones with these human cell lines ranged between 33-46% indicating that Hop is not absolutely essential in human cells similarly to what had previously been found for the budding yeast *Saccharomyces cerevisiae*[32]. The Hsp90α and Hsp90β KO cells will be described in more detail elsewhere.

**Flow cytometry**. For all flow cytometric analyses, a minimum of 10,000 cells were analyzed for each sample. We used a FACSCaliber (BD Biosciences) with the software package CellQuest Pro and a FACS Gallios (Beckman Coulter), and data were analyzed by with the software packages FlowJo and CellQuest Pro. Additional details are given in the following paragraphs. Gating and data analysis strategies are presented in Supplementary Figs. 11 and 12.

*Annexin V-PI staining*. Human cells seeded at a density of $5 \times 10^5$ per 2 ml were harvested by trypsinization, washed in phosphate-buffered saline (PBS) and resuspended in 100 μl annexin V-binding buffer (10 mM HEPES pH 7.4, 150 mM NaCl, 2.5 mM CaCl₂). Annexin V-FITC (5 μl) and propidium iodide (PI; 2.5 μg/ml) were added to the cells and incubated for 15–20 min at 4 °C. In the dot plot analyses, the first quadrant represented healthy, unstained cells; the second quadrant represented early apoptotic cells with only annexin V-FITC staining; the third quadrant represented late apoptotic cells with both annexin V-FITC and PI staining; and the fourth quadrant represented necrotic cells with PI staining.

*Cell death assays*. Human cells were resuspended in 100–200 μl of PBS-containing PI (2.5 μg/ml) for 15–20 min at room temperature (RT) before flow cytometric analysis.

*Cell cycle analyses*. Cells were harvested as outlined above. Cells were fixed with 70% ice-cold ethanol, washed, treated with 100 μg/ml RNase A at RT for 5 min, then incubated with 50 μg/ml PI for 15–20 min at RT before flow cytometric analysis. Apoptotic cells were identified by the quantitation of the SubG0 cell population.

*Autophagic flux measurement*. Cells seeded at a density of $4 \times 10^5$ per 2 ml were transfected with the autophagy reporter plasmid FUW mCherry-GFP-LC3 using PEI. At 48 h post-transfection, cells were harvested by trypsinization, and GFP- and mCherry-positive cells were measured by flow cytometry. The autophagic flux was measured by calculating the ratio of the mean fluorescence intensity of GFP- and mCherry-positive cells. A lower relative ratio of GFP/mCherry is indicative of a higher autophagic flux.

**Cell proliferation assay**. Human cells seeded at a density of $4 \times 10^4$ per 200 μl were grown for 24, 48 or 72 h after which, 100 μg/ml MTT (3-(4,5-dimethylthiazol-2-yl)-2,5-diphenyltetrazolium bromide;) was added to the culture medium for 3 h. The culture medium was removed, the formazan crystals were dissolved in DMSO, and the OD measured at 550 nm with a plate reader (Tecan Sunrise).

**Biochemical fractionation of soluble and insoluble proteins**. To analyze the polyglutamine (polyQ) protein aggregation within cells, cells seeded at a density of $7 \times 10^5$ per 2 ml were transfected with plasmids pEGFP-Q74 and pEGFP-Q23 with PEI. At 48 h post-transfection, cells were harvested for biochemical fractionation. For some experiments, at 24 h post-transfection, cells were treated with GA (0-500 nM) for 24 h. For the comparative analysis between WT and KO cells, soluble and insoluble fractions were biochemically separated from $4 \times 10^6$ cells for each genotype as described previously[88]. Briefly, cells were lysed in a lysis buffer with mild detergents (20 mM Tris-HCl pH 7.4, 2 mM EDTA, 150 mM NaCl, 1.2% sodium deoxycholate, 1.2% Triton-X-100, 200 mM iodoacetamide, protease inhibitor cocktail [PIC], sonicated (low power, three cycles of 10 s pulses), and centrifuged at $16,100 \times g$ for 20 min. The supernatant was collected as the soluble fraction. The precipitate (insoluble fraction) was washed 5–6 times with PBS and solubilized in NuPAGE protein sample buffer (Life Technologies, cat no. NP0008) complemented with 10 mM DTT. Both biochemical fractions were analyzed by immunoblotting.

**In vitro proteasomal activity assay**. Human cells were harvested and washed. Cell pellets were resuspended in lysis buffer (25 mM Tris-HCl pH 7.4, 250 mM sucrose, 5 mM MgCl₂, 1% NP-40, 1 mM DTT, 1 mM ATP) and incubated for 10–15 min on ice. Samples were centrifuged at $16,100 \times g$ for 20 min and supernatants were collected for the proteasomal activity assay. Yeast cells were collected, washed with H₂O and lysed mechanically with glass beads ($3 \times 30$ s pulses) in lysis buffer (10 mM Tris-HCl pH 7.4, 50 mM NaCl, 10 mM MgCl₂, 1 mM EDTA, 20% glycerol, 1 mM DTT, 1 mM ATP). Samples were centrifuged at $16,100 \times g$ for 20 min and supernatants were collected for the proteasomal activity assay. Equal amounts of protein (25–50 μg) for each sample was diluted in proteasomal reaction buffer (50 mM Tris-HCl pH 7.4, 5 mM MgCl₂, 1 mM DTT, 1 mM ATP) in a 96-well opaque bottom white plate and 50 μM *N*-succinyl-Leu-Leu-Val-Tyr-7-amino-4-methylcoumarin (suc-LLVY-AMC) was added to each well. AMC fluorescence

was measured at 460 nm for 5–60 min, with an excitation at 380 nm. Purified proteasomes (1 μg) were diluted in proteasomal reaction buffer and activity was measured with suc-LLVY-AMC. Alternatively, the Proteasome Activity Assay Kit (Abcam) was used according to the manufacturer's instructions to measure proteasomal activity. All fluorescence measurements were recorded using a plate reader (Cytation 3, BioTek).

**In vivo UPS activity assay**. Cells seeded at a density of $5 \times 10^5$ per 2 ml were transfected with plasmids Ub-M-GFP (stable GFP) and Ub-R-GFP (degradation-prone GFP) with PEI. At 48 h post-transfection, cells were harvested by trypsinization and GFP-positive cells were quantitated by flow cytometry[38]. The in vivo UPS activity was expressed as the % Ub-M-GFP-positive cells minus the % Ub-R-GFP-positive cells.

**26S/30S proteasome purification from human cells**. Proteasome particles were purified from mammalian cells as described previously[89]. Human cells were cultured in 15 cm cell culture dishes until 85–90% confluency was reached. Cells were harvested, resuspended in lysis buffer (25 mM HEPES-KOH pH 7.4, 40 mM KCl, 5 mM MgCl₂, 10% glycerol, 1 mM DTT, 1 mM ATP) and lysed by sonication (three cycles of 30 s pulses). The lysates were clarified by centrifugation and the supernatants were filtered through 0.45 μm membranes. Clarified lysates were incubated with 1 mg GST-UBL (final concentration of 0.1–0.2 mg/ml) and GSH-agarose beads. The mixture was loaded onto a column and washed with 40 column volumes of lysis buffer. The proteasome was eluted in two rounds with 250 μl of 10× His-tagged UIM (≥2 mg/ml) with a 15 min incubation before elution at 4 °C. The eluted proteasomal fraction was further applied on Ni-IDA affinity matrix to remove excess 10× His-tagged UIM. Purified proteasomes were stored at −80 °C.

**In vitro luciferase refolding assay**. The luciferase refolding assay was adapted from a previous publication[61]. Specifically, to test the importance of Hop in the refolding assay, luciferase from the firefly *Photinus pyralis* was diluted to 100 nM in refolding buffer (25 mM HEPES-KOH pH 7.6, 100 mM KOAc, 10 mM Mg(OAc)₂, 2 mM ATP, 5 mM DTT) containing 1 μM Hsp70, DnaJB1 and Apg2 (Hsp70: DnaJB1:Apg2; 2:1:0.5) and 1 μM Hsp90α, when indicated. Recombinant human Hop was added in the indicated experimental sets in different concentrations (Hsp90α:Hop = 1:0.25, 1:0.5, 1:1). Luciferase was heat-denatured at 42 °C for 10 min and refolding was allowed at 30 °C for the indicated times. To test the Hsp90α double mutant K418A, K419A, luciferase was diluted to 80 nM in the refolding buffer in the presence of Hsp70 (4 μM), DnaJB1 (0.5 μM), Apg2 (0.25 μM), along with either WT (0.5, 1 or 2 μM) or mutant Hsp90α (0.5, 1, or 2 μM). In this case, luciferase was allowed to refold for 49, 59, and 69 min and the refolding values of these three time points were averaged (for triplicate samples). Luciferase activity of the reaction mixture was measured in assay buffer (100 mM K-phosphate buffer pH 7.6, 25 mM glycylglycine, 100 mM KOAc, 15 mM Mg(OAc)₂, 5 mM ATP) by adding the substrate luciferin (final concentration 80-100 μM). Measurements were made with a luminometer plate reader (Berthold Technologies, XS3 LB930 with Mikrowin Software 2010 or SpectraMax iD3 Multi-Mode Microplate Reader of Molecular Devices with Softmax Pro 7 Software).

**In vivo luciferase refolding assay**

*Human cells*. Cells ($2 \times 10^4$) transfected with the luciferase expression vector pC7L were resuspended in 100 μl of cell culture medium. Cells underwent HS at 42–43 °C for 15 min to denature luciferase, followed by incubation at 37 °C for 1–3 h for the refolding of luciferase. Cells were harvested by centrifugation and lysed with Passive Lysis Buffer (Promega). Cell extract (10 μl) was mixed with an equal volume of firefly luciferase assay substrate from the Dual-Luciferase detection kit (Promega) and the luciferase luminescence signals were measured by the Chameleon bioluminescence plate reader (Noki tech.). To determine the impacts of Hsp70 and Hsp90 inhibitors on luciferase refolding, 2 μM of JG-98 or 1 μM of GA were added to pC7L-transfected cells 1 h prior to HS. HS and luciferase refolding were performed in complete cell culture medium containing the same concentrations of JG-98 and GA. The luciferase activity of cells not subjected to HS was set to 100%. Alternatively, cells were co-transfected with pC7L and either WT or TPR mutant Hop. At 48 h post-transfection, a luciferase refolding assay was performed as described above except that luciferase was heat-denatured at 44 °C for 20 min.

*Yeast cells*. pLG/Luc transformants were grown O/N in galactose-containing minimal medium at 30 °C. Cells were diluted and grown to mid-log phase ($OD_{600} = 0.6–0.7$), washed with water and resuspended in minimal medium containing glucose and 100 μg/ml CHX; $OD_{600}$ was adjusted to 0.4 for each sample. 100 μl samples of yeast cells were subjected to HS at 44 °C for 10–30 min to heat-denature luciferase, the refolding of luciferase was allowed by incubation at 30 °C for 1–3 h. Whole-cell luciferase activity was measured by mixing 50 μl of processed yeast cells with equal volumes of 100 mM Na-citrate buffer (pH 5.0) and 1 mM D-luciferin, before luminescence was measured with a bioluminescence plate reader. The luciferase activity of cells not subjected to HS was set to 100%.

**Immunoprecipitation (IP)**. Hsp90α, Hsp90β, Hop, and FLAG-Hsp90α IP: Human cells were resuspended in Hsp90 complex lysis buffer (10 mM Tris-HCl pH 7.5, 50 mM NaCl, 1 mM EDTA, 1 mM DTT, 10% glycerol, 10 mM Na-molybdate, 0.01% Triton X-100, PIC) and lysed by sonication (30-40 cycles of 30 s) using a Bioruptor® sonicator (Diagenode). For Hsp90α or Hsp90β IP, 1 mg of clarified cell extract was mixed with 5 μg anti-Hsp90α (9D2) or anti-Hsp90β (H90-10) antibodies. For FLAG-Hsp90α IP, ~1 mg of cell extract was mixed with 10 μg anti-FLAG (M2) antibodies. For Hop IP to detect associated proteasomal components, 2–3 mg of cell extract was mixed with 5 μg anti-Hop antibodies.

For all experiments, IPs were incubated O/N at 4 °C on a rotating wheel before 50 μl of Dynabeads™-Protein G (Thermo Fisher Scientific) were added and incubated for 3 h at 4 °C. The Dynabeads were washed and boiled with the NuPAGE protein sample buffer containing 10 mM DTT. The eluates were collected, separated by SDS-PAGE (7.5–10%) and visualized by immunoblotting.

**Native-PAGE analysis**. For experiments to visualize proteasomal complexes in total cell lysates, human and yeast cells were lysed in 25 mM HEPES-KOH pH 7.4, 40 mM KCl, 5 mM MgCl$_2$, 10% glycerol, 1 mM DTT, 1 mM ATP, PIC at 4 °C. Lysis was done by sonication with 30 cycles of 30 s on/off with a water bath sonicator and by shaking with glass beads with three 30 s pulses, respectively. Equal amounts of total cell extracts (40–75 μg) or purified proteasomes (2 μg) were separated by 4% Tris-HCl native-PAGE. For all samples, native-PAGE was performed for 3.5–4 h at 100–110 V in a 4 °C cold room. Separated proteins were transferred onto nitrocellulose membrane and visualized with proteasomal subunit-specific primary antibodies.

**In vitro protein–protein interaction assay**. Five micrograms of purified WT or point mutant Hsp90α or WT Hsp90β protein were mixed with 7.5 μg of purified Hsp70 (V438F) in association buffer (10 mM Tris-HCl pH 7.5, 50 mM NaCl, 1 mM EDTA, 1 mM DTT, 0.01% Triton-X-100, 10% glycerol, 10 mM Na-molybdate, PIC) in the presence or absence of 5 mM ATP. The mixtures were incubated for 1–2 h at 30 °C and the association of Hsp90 and Hsp70 was determined by a co-IP. Briefly, 5 μg anti-Hsp90α (9D2) or anti-Hsp90β (H90-10) antibodies along with 25 μl of Dynabeads™-Protein G were added to the protein mixtures and incubated O/N at 4 °C. Equal amounts of normal IgG were used for the corresponding control IP. Dynabeads were washed and boiled with the NuPAGE protein sample buffer complemented with 10 mM DTT, and eluates were collected. Further SDS-PAGE and immunoblotting were performed with eluted protein samples, using purified proteins as inputs.

**Assay of protein translation rate**. Cells seeded at a density of $5 \times 10^6$ per 10 ml were treated with puromycin (1 μM) for 0–2 h, harvested and lysed in 20 mM Tris-HCl pH 7.4, 2 mM EDTA, 150 mM NaCl, 1% sodium deoxycholate, 1.2% Triton-X-100, 200 mM iodoacetamide, PIC. For immunoblotting, 50 μg of clarified cell lysates were probed with anti-puromycin antibodies to reveal the newly synthesized proteins or polypeptides. In this assay, the rate of incorporation of puromycin into newly synthesized proteins is directly proportional to the global rate of translation.

**Immunoblot analyses**. Lysates of cells (20–100 μg) or purified proteins/proteasome (1–5 μg) were subjected to SDS-PAGE/native-PAGE and transferred to a nitrocellulose membrane (GVS Life Science) with a wet blot transfer system (VWR). Membranes were blocked with 2–5% non-fat milk or BSA in TBS-Tween 20 (0.2%) and incubated with primary antibodies (Supplementary Data 5) with the following dilutions: anti-Hop (1:1000), anti-GAPDH (1:7500), anti-HA-tag (1:5000), anti-FLAG tag (1:5000), anti-His tag (1:5000), anti-Hsp70 (1:2000), anti-Hsc70 (1:2000), anti-Raf1 (1:1000), anti-Psmd1 (1:1000), anti-Psmd2 (1:1000), anti-Psmd6 (1:1500), anti-Psmd9 (1:750), anti-Psmd5 (1:750), anti-Paaf1 (1:750), anti-Psmc5 (1:2000), anti-α-tubulin (1:5000), anti-Psma3 (1:2000), anti-20S CP (1:1000), anti-Ub (1:5000), anti-EGFP (1:7500), anti-p-Tyr (1:2000), anti-p-Ser (1:1000), anti-Hsp90α (1:1000 and 1:2000 for the antibodies from Thermo Fisher Scientific and Enzo Lifesciences, respectively), anti-Hsp90β (1:2000), anti-p23 (1:1000), anti-Cdc37 (1:1000), anti-Akt (1:1000), anti-Cdk2 (1:500), anti-Erbb4 (1:1000), anti-Hsf1 (1:1000), anti-p53 (1:1000), anti-β-catenin (1:1500), anti-Stat3 (1:1500), anti-BclXl (1:1000), anti-Bcl2 (1:1000), anti-HDAC6 (1:1000), anti-vimentin (1:500), anti-v-Src (1:1000), anti-progesterone receptor (1:1000), anti-glucocorticoid receptor (1:1000), anti-androgen receptor (1:1000), anti-Sti1 (1:750), anti-Cdk4 (1:1000), anti-p-c-Src (Y416) (1:1000), anti-c-Src (1:1000), anti-H2A.Z (1:1000), anti-p-Erk (1:500), anti-Erk1/2 (1:1000), anti-Hsp40/Hdj1 (1:1000), anti-Hsp110 (1:1000), anti-Puromycin (1:22000), anti-Rpt1 (1:1000), and anti-Rpt2 (1:20000). Membranes were washed with TBS-Tween 20 (0.2%) and incubated with the corresponding secondary antibodies: anti-rat IgG-HRP (1:10000), anti-mouse IgG-HRP (1:10000), anti-rabbit IgG-HRP (1:10000), and anti-mouse IgM-HRP (1:10000). Immunoblots were developed using the WesternBright™ chemiluminescent substrate (Advansta). Images were recorded by using X-ray film (Fujifilm) or with a LI-COR Odyssey image recorder.

**Mass spectrometry**

*General LC-MS/MS analysis*. Samples were analyzed on a Fusion orbitrap trihybrid mass spectrometer, interfaced via a nanospray source to a Dionex RSLC 3000 nano

HPLC system (Thermo Fisher Scientific, Bremen, Germany). Extracted peptide mixtures were separated on a custom-packed reversed-phase C18 nanocolumn (75 μm ID ×40 cm, 1.8 μm particles, Reprosil Pur, Dr. Maisch) with a gradient from 5 to 55% acetonitrile in 0.1% formic acid for 120 min (for IP samples: same gradient in 35 min). Full MS survey scans were performed at 120,000 resolution. All survey scans were internally calibrated using the 445.1200 background ion mass. Using a data-dependent acquisition controlled by Xcalibur 2.1 software (Thermo Fisher), a maximum number of multi-charged precursor ions was selected for tandem MS analysis within a maximum cycle time of 3 s. Selected precursor ions were fragmented by Collision-Induced Dissociation (CID) and analyzed in the linear ion trap with an isolation window of 1.6 $m/z$. Selected ions were then dynamically excluded from further selection during 60 s.

**Phase contrast and fluorescence microscopy**. Cellular morphology was analyzed using an inverted light microscope (Olympus CK2) and phase contrast images were captured with a Dino-lite camera using the DinoXcope software. Cells were seeded on glass coverslips and transfected with the plasmids Q74-EGFP and Q23-EGFP. At 48 h post-transfection, cells were fixed with 4% paraformaldehyde and mounted on glass slides using Mowiol. EGFP-positive cells were visualized and images were captured with a fluorescence microscope (Zeiss, Germany) at ×20 magnification.

**Negative staining transmission electron microscopy**. Carbon-coated copper grids (200 or 300 mesh) were glow discharged for 20 s. Purified proteasomes were diluted in buffer (25 mM HEPES-KOH pH 7.4, 40 mM KCl, 10% [v/v] glycerol, 5 mM MgCl$_2$, 1 mM ATP, 1 mM DTT) and 5 μl of samples were loaded onto the grid surface and incubated for 30 s. Excess samples were blotted off using filter paper while holding the grid vertically. Two drops of (100 μl each) uranyl acetate (2% in water) were prepared for each grid. Grids were incubated for 2 s on a uranyl acetate drop and on a second drop for 30 s to stain the grids. Processed grids were analyzed with a transmission electron microscope at 120 kV (Tecnai G2, FEI, Eindhoven, Netherlands).

**Confocal microscopy**. Hop WT and KO A549 cells were fixed with 4% formaldehyde and subsequently permeabilized with 0.1% Triton X-100. Cells were blocked with 1% bovine serum albumin and incubated with primary antibodies against Psma3, Psmc5, and Psmd6 (all at 1:200 dilution) O/N in a moist chamber at 4 °C. Appropriate AlexaFluor488-conjugated secondary antibodies were used at a 1:2000 dilution at RT for 1 h. After DAPI staining, cells were mounted with mowiol and data acquisition was done with a Leica SP8 confocal microscope with a 63× oil immersion objective. Digital images were acquired with 2× digital magnification. Images were analyzed with the software ImageJ-Fiji.

**Transcriptional activity assays**. Human cells were seeded at a density of $6 \times 10^4$ per 0.5 ml in phenol red-free DMEM medium containing 5% charcoal-treated FBS in 24-well cell culture plates. The cells were co-transfected with ERα, GR or PR mammalian expression plasmids (only for HEK293T cells), as well as both a luciferase reporter plasmid and a constitutive Renilla luciferase expression plasmid (pRL-CMV). In another experiment, cells were transfected with luciferase reporter plasmids as described for Supplementary Fig. 9d along with pRL-CMV. At 48 h post-transfection, cells were lysed with Passive Lysis Buffer, and firefly and Renilla luciferase activities were measured using the Dual-Luciferase detection kit (Promega) with a bioluminescence plate reader. Renilla luciferase activity was used as transfection control.

**RNA extraction and quantitative reverse transcription PCR (qRT-PCR)**. Cells were lysed with the guanidium-acid-phenol method by using the TRI reagent (4 M guanidium thiocyanate, 25 mM sodium citrate, 0.5% N-lauroylsarcosine, 0.1 M 2-mercaptoethanol, pH 7). Then sequentially 2 M sodium acetate pH 4, aquaphenol and chloroform:isoamyl alcohol (49:1) were added to the lysates and mixed. Organic and aqueous phases were separated by centrifugation at $10,000 \times g$ for 20 min and the aqueous phases were collected. RNA was precipitated by adding isopropanol and centrifugation at $12,000 \times g$ for 20 min at 4 °C. RNA pellets were washed with 75% ethanol and resuspended in nuclease-free water. cDNA was prepared from RNA (400 ng) by using random primers (Promega), GoScript buffer (Promega), and reverse transcriptase (Promega) according to the manufacturer's instructions. cDNAs were mixed with the GoTaq master mix (Promega) and specific primer pairs for *STIP1* (Supplementary Data 5) for quantitative PCR with a Biorad CFX96 thermocycler. Expression of mRNA of the gene of interest was normalized with *GAPDH* as internal standard.

**Bioinformatic analyses**

*In silico protein homology modeling*. Beginning with the apo conformation of HtpG (PDB ID: 2IOQ)[90], homology models of the human Hsp90α and Hsp90β proteins were constructed using the I-TASSER structure prediction web server[91]. Briefly, monomeric structures were predicted according to the C-score and TM-score of Hsp90α (C-score = −1.53, TM-score = 0.53 ± 0.15) and Hsp90β (C-score = −0.99, TM-score = 0.59 ± 0.14). To generate dimer structures of Hsp90α and Hsp90β, full-length predicted monomeric structures of each protein were manipulated using

the ClusPro protein–protein docking web server programmed in dimer mode[92]. Hydrophobic interaction driven dimer structures, with low free energy, were selected. The accessibility of surface residues of the dimeric Hsp90α and Hsp90β were predicted with the Swiss-PdbViewer 4.1.1 software. Note that our modeling and the surface-exposure of the highlighted residues are supported by a recent study of Tastan Bishop and colleagues[93].

*Gene ontology enrichment analysis.* GO enrichment analyses with the whole-cell MS datasets were done with those proteins, which changed the most (fold change >0.4 or smaller than −0.4 and with a VIP value over (1) with the ClusterProfiler package for R. Overrepresentation of "Biological Process" GO terms in the data of Hop KO cells was determined by using the database "org.Hs.eg.db" from Bioconductor. GO enrichment analyses for the Hop (HA)-IP-MS dataset were performed with the Enricher web server (http://amp.pharm.mssm.edu/Enrichr), focusing on KEGG and WIKI pathway-annotated proteins. Significant biological processes were identified by lower *p*-values, a higher number of common genes/proteins responsible for a biological process, and a high combined score provided by the Enricher web server.

*Protein interaction network analysis.* Fold change values (WT vs. KO) of proteins from the whole-cell MS analyses were analyzed as described previously[94]. All nodes correspond to identified proteins. Cytoscape analyses were represented as heat maps of a log2 fold change of protein expression. Hsp90-specific clients, chaperones and co-chaperones were classified using our own web server for Hsp90 interactors (https://www.picard.ch/Hsp90Int/index.php).

**General data analyses**. Data processing and analyses were performed using GraphPad Prism (versions 7 and 8).

**Mass spectrometric data analyses**. Tandem MS data were processed by the MaxQuant software (ver.1.6.0.13)[95] incorporating the Andromeda search engine[96]. The UniProt human reference proteome database (of October 2017) was used (71,803 sequences), supplemented with sequences of common contaminants. All identifications were filtered at 1% false discovery rate at both the peptide and protein levels with default MaxQuant parameters. For the whole-cell proteome and Hsp90 IP-MS experiments, from the relevant MS datasets only protein groups were kept for the subsequent analyses, which met the criteria of at least two unique peptides, four peptides, and six MS/MS counts. Protein quantitation was performed either with the iBAQ values (for the IP-MS and proteasomal analyses) or with the LFQ values (for the whole-cell proteome analyses)[97]. In whole-cell proteome and Hsp90 IP-MS analyses, the log2 fold change >0.4 or <−0.4 of a protein was considered as the biologically significant difference. Proteins that were detected in at least one of three biological replicates (for the whole-cell proteome) were used in all further analyses.

In Hsp90 IP-MS analyses, the criteria for the validation of interactors was an average fold change of 8 (3 in log2 scale) or more, identified by MS/MS in all positive IP samples compared to control IP, and an adjusted *p*-value < 0.05. The iBAQ values of interactors were normalized on the sum of the iBAQ of Hsp90β and Hsp90α.

In Hop (HA) IP-MS analyses, the interactor validation criteria was an average fold change of 40 or more identified by MS/MS in WT Hop IP samples compared to the control IP and at least 3 unique peptides. Since chaperone, co-chaperone and proteasomal proteins were our focus, we included three proteasomal components (PSMB1, PSMD7, and PSMD14) and three molecular chaperones (HSP90AB4P, HSP6, and HSP7) in further analyses despite them being represented by only 2 unique peptides. The iBAQ values of interactors were normalized either on the abundance of WT or TPR mutant Hop depending on the experimental set. PCA and (O)PLS-DA analyses of the results from whole-cell proteome and Hsp90 IP-MS experiments were done using the ropls package for R[98]. The variable importance in projection (VIP) values were derived from the "Orthogonal Projections to Latent Structures Discriminant Analysis" (OPLS-DA) model and used to select variables of interest with a VIP cut-off of 1[99].

**Proteasomal activity analyses**. For steady-state proteasomal activity measurements, AMC fluorescence of WT samples for each time point was considered as 100% proteasomal activity and the corresponding proteasomal activity of KO samples was calculated relative to that. For the proteasomal activity rate measurements, AMC fluorescence of each time point was normalized with AMC fluorescence at 5 min; data were represented as the fold change while slopes were considered as the rate of proteasomal activity over time.

**TEM image analyses**. Eight-hundred twenty-seven and 705 micrographs were acquired for proteasome samples from Hop WT and KO HEK293T cells, respectively, at 2.245 Å/pix using an Eagle detector (Thermofisher). The micrographs were analyzed using Relion (version 3.1)[100]. 106,491 and 92,102 particles were picked for the WT and KO samples, respectively, using Gaussian picking implemented in Relion. The particles were initially classified for four rounds. For the WT sample, 17,342 and 7314 particles were selected for the top and side views, respectively, and for the KO sample, 20,120 and 4410 particles were selected for the top and side views, respectively, and each classified separately for another two rounds. Finally, the particles of good classes corresponding to each view and sample were selected and statistically analyzed.

**Cell death analysis**. The calculations of the percent dead cells for the experiments of Supplementary Fig. 7f, g were performed as follows. First, the background of dead cells in the controls were subtracted from the corresponding MG132-treated sets for both 37 °C and HS-treated cells. Then the percent dead cells at 37 °C of a particular genotype was subtracted from the percent dead cells of the same genotype under HS treatment and the final measurement was considered the percent cell death under the experimental condition of HS + MG132. This explains how some cells or some conditions can yield negative values.

**Reporting summary**. Further information on research design is available in the Nature Research Reporting Summary linked to this article.

## Data availability

All data supporting the findings of this study are available from the corresponding authors upon reasonable request. The mass spectrometric proteomic data are available through the ProteomeXchange Consortium with the identifier PXD012774 and for a subset in Supplementary Data 1–4. Source data of uncropped immunoblot images and the individual data points are provided with this paper. Source data are provided with this paper.

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

## Acknowledgements

We are grateful to Alfred L. Goldberg, Matthias P. Mayer, Ueli Schibler, Marcello Maggiolini, Donald McDonnell, Yoshihiko Miyata and Adrienne Edkins for gifts of plasmids. We thank David O. Toft, and Martine Collart for gifts of antibodies, Jason Gestwicki for the gift of JG-98, Irene Olivan Mura for contributing to the in vitro refolding experiments, Stacey Mattison for critically reading the Methods section of this manuscript, and previous members of the Picard laboratory for miscellaneous reagents. This work has been supported by the Swiss National Science Foundation and the Canton de Genève.

## Author contributions

K.B. conceived the study, designed and performed experiments, analyzed the data, prepared figures, and wrote the manuscript. L.W. and M.Q. conducted the proteomic analyses. T.M.L, E.C.P., M.B.K., and S.G.D.R. designed and performed in vitro luciferase refolding assays. P.C.E. performed bioinformatics analyses of proteomic datasets. M.V. generated Hsp90α- and Hsp90β-KO cells. L.B. contributed to experiments with recombinant proteins. D.W. helped with all yeast experiments. Y.S. and C.B. contributed to TEM experiments and analyses. D.P. conceived the study, contributed to designing the experiments and analyzing the data, supervised the work, and wrote and edited the manuscript. All authors provided critical analysis of the data and contributed to the editing of the manuscript.

## Competing interests

The authors declare no competing interests.
