## [Peer Review File · Nature Communications]

REVIEWER COMMENTS

Reviewer #1 (Remarks to the Author):

In this study the authors attempt to characterize the role of the Hsp90 co-chaperone Hop by using a clever series of experiments/techniques including wide-scale proteome and interactome analysis as well as construction of several useful Indel CRISPR cell lines. From their analysis, they focus their attention on the proteasome system which is functionally impacted by loss of Hop. Despite a good premise, this work does not go far enough to characterize the molecular nature of the defect in proteasomal function. Many of the figures feel crowded with negative/control results that could be put in supplemental. Although the authors did a fantastic job with their proteomics analysis, they do not really explore the data in enough detail. Specific comments:

1) The foundation of this work rests on utilization of CRISPR KO cell lines. As the authors are aware, this technology does not produce clean knockouts as with micro-organisms. Several non-ideal situations can arise. First, a truncation of the protein can be produced (from insertion of a premature stop codon). Secondly, a frame shift mutation can occur which leaves a small part of the original protein and then after the frame shift a random amino acid sequence. Although the authors attempt address this by Western Blot (Fig. 1) and by analyzing peptides from WT and KO cells (Table S1), this is not a good enough validation. The genomic area around the gRNA binding site needs to be sequenced to fully characterize where the stop codon was introduced. Please note, this needs to be done for both copies of the genes involved. For more details please see <https://www.nature.com/articles/nprot.2013.143>

2)Figure 1: The authors spend a large amount of figure real estate on showing little difference between WT and KO cells (Fig. 1a-h). While important, the truly interesting data is the omic data in 1i. While it is clear that the majority of the proteome remains unchanged, there are proteins significantly altered in abundance. I recommend moving some of the earlier "control" figures 1a-h to supplemental, then making 1i larger with GO analysis on the most changed proteins. Correspondingly, there should be more analysis of 1i in the text.

3)Figure 3: The analysis of the Hop interactome is very exciting, but could be represented in a clearer manner:

-3a is really a control experiment and should be in supplemental. Instead, I recommend a schematic of the proteomics workflow for fig. 3a.

-The pie chart in 1b is not clearly worded, please change to "1.57%, 10 enriched in K8A, K298A Hop".

-the notation for the double mutant seems odd. I would revise to either "AA Hop" or "K8A, K298A Hop".

-3c-the y-axis legend seems to have an extra "w" in it.

-3c This seems like an ineffective way to represent such nice data. At the very least I would change the y-axis to a log scale or even better, draw the network of interactions and color them based on enrichment.

-Given that the majority of the interactions go away in the mutant Hop that cannot bind Hsp90, are the authors detecting simply Hsp90 interactors rather than direct interactors of Hop? Rephrased another way, are you detecting Hsp90-client complexes present when Hop is bound to Hsp90? There needs to be some clarity in the text on this.

-Some of the important interactions detected in the MS experiment should be validated in vivo (with IP-Western Blots) from both HEK cells and in vitro (IP-Western with recombinant versions of the proteins) to ascertain direct vs bridged interactions.

4) Figure S3K. The authors mentions KO of Hsp90a and b. Given that there is disagreement in the literature on the essential nature of Hsp90 isoforms, the authors need to sequence their KO cells as in comment 1.

5) Figure 4: As with many of the figures, Figure 4 seems unnecessarily crowded with control/negative results. An example is 4c, which is not showing any impact of the knockout on proteasomal components. Readers can simply refer to the data in a supplementary table. The key take-home message of this figure is that proteome capping may be altered in Figure 4e, but is not convincing.

6) Overall, we suggest that the authors look at possibly re-arranging figures 3 and 4. Figure 3 should really be a clear analysis of the Hop interactome (Hsp90-dependent and independent) and Figure 4 should focus on the proteasome. At the moment, the foci of these figures is blurred and overlapping.

7) The authors try to examine the exact nature of the proteasomal defect. They clearly show steady state levels of the proteasomal subunits are not altered and show unconvincing data on the formation of the proteasome complex. There are other potential ways the proteasome could be impacted. For example, are PTMs on the individual proteasomal subunits altered in cells lacking Hop? The protein levels may be unchanged but PTMs may be different. This can easily be examined by the kind of proteomics the authors demonstrate in the paper.

8) Alternatively, the subunits may assemble fine in vitro but not in vivo. Is it possible that the localization of the proteasomal subunits is altered in Hop KO cells?

Reviewer #2 (Remarks to the Author):

In this paper, the authors address the role of Hop, a Hsp90 and Hsp70 cochaperone, in proteostasis. They present data showing that Hop knockout cells are unable to efficiently cap the core particles of the proteasome with regulatory particles and thus are inefficient at degradation compared to WT cells. They further show that knockout cells are better able to prevent aggregation and promote protein folding than WT cells. The results suggest that the one role of Hop is to shift the proteostatic balance from refolding towards degradation.

A great deal of work has gone into this project and the conclusions are supported by the data presented. The results and conclusions are exciting and will be of interest to specialists in the field of chaperones as well as to a general audience. However, one major comment is that it is not written for a general audience, but rather for specialists in the field. The impact of the most important conclusions is diminished by being mingled with the other observations, negative results and less important conclusions and suggestions.

Comments:

1. Overall the paper could be significantly improved by writing in a clearer, more concise manner with the emphasis on the major findings. For example, the first section could go in the supplementary information or be moved to a less prominent place in the manuscript, since much of the data is negative and might have been expected from previous yeast results. The conclusion could be stated with a reference to the SI.

2. Overall the figures could be clearer. Overall, many of the panels would be easier to follow if they were better labeled. Without looking up instructions to authors, the figures appear to have too many panels per figure. Figs. 1, 2, 3, 4, 5 and 6 have so many panels that it is difficult (impossible) to read the words when printed on paper.

3. In general, figure legends could be improved by adding necessary information, so the reader understands what is shown. The reader has to go back to the methods to find out what components are in the experiments and how many replicates were performed.

4. For many experiments, data with two knockout strains are presented. It is important to show that there is not just one unique knockout. However, it might be easier for the reader if some of the data for the two strains were put in the SI and if the figures had fewer panels. The same applies for the

yeast strains.

5. Perhaps the wording of the last sentence of the abstract could be improved. "Thus, cells may act on Hop to shift the proteostatic balance between folding and degradation." It is not that cells are acting on Hop, but rather regulatory proteins.

6. The meaning of "axis" in the section heading for section 2 is not clear.

7. Fig. 1a. Move Hop to line up with band.

8. The cartoon in Fig. 2A is confusing. How can aggregates of "misfolded proteins" get converted to "disaggregated peptides". One choice is to change "peptide" to "proteins", since peptides are less than 50 amino acids. However, how are "disaggregated proteins" different than "misfolded proteins". The cartoon shows that they are. It might be clearer if the arrow from aggregates went back to the pool of "misfolded proteins" and that pool was referred to as "unfolded and misfolded proteins." Since it is called "a proteostatic equilibrium" in the legend, perhaps some arrows should go in both directions.

9. Fig. 3b. Label as "Hop interactors". Check significance of numbers (1.57%). Eliminate the number for each category, since the total number is given.

10. Fig. 3d. The pie graph is not necessary.

11. Fig. 3e, f and h. Not every KO strain needs to be shown and the rest could go in SI.

12. Fig. 4a, change title to something like "fold change (log2) following incubation with whatever Hsp90 inhibitor was used" (it is not mentioned in the text or the fig. legend).

13. Fig. 4b, c, d could go in SI.

14. Fig. 4e. It is confusing what the difference is between capped and 30S particles? Are the two graphs showing different things?

15. Fig. 4g. One strain could be moved to the SI.

16. Fig. 5a, k and l could be moved to the SI.

17. Fig. 6a. Cartoon is not clear. Perhaps show the pathway consistent with the results and leave out the "expectations" since they were not consistent with the data. It would be helpful to label the cartoon for clarity (90, 70, Hop, J).

18. Fig. 7, g and h cannot be compared since different concentrations of proteins were used and the data are plotted differently.

Sue Wickner

Reviewer #3 (Remarks to the Author):

I was asked to review the structural and bioinformatics aspects of this manuscript. I read through the whole article though, to form a general opinion. In general, I found that the manuscript is very clearly and logically written. The scientific questions are well put and the experiments are well designed.

In my opinion the article is acceptable subject to the following corrections:

1. In Introduction section, authors state that "Hop forms a ternary complex with Hsp70 and Hsp90 using its tetratricopeptide repeat (TPR) domains. Two of its three TPRs, TPR1 and TPR2A, specifically bind the extreme C-terminal sequences EEVD and MEEVD of Hsp70 and Hsp90, respectively." However, the literature reports other potential interactions between Hop and Hsps; including Hop-Hsp90 middle domain interactions. Hence, authors should cover all the potential interactions between these proteins as reported previously. Thus, their references need to include at least these two articles:

A.B. Schmid, S. Lagleder, M.A. Gräwert, A. Röhl, F. Hagn, S.K. Wandinger, et al., The architecture of functional modules in the Hsp90 co-chaperone Sti1/Hop, *EMBO J.* 31 (2012) 1506–1517.

Hatherley R, Clitheroe CL, Faya N, Tastan Bishop Ö. Plasmodium falciparum Hop: detailed analysis on complex formation with Hsp70 and Hsp90. *Biochem Biophys Res Commun.* 2015;456(1):440-445.

2. Authors indicate that "We did notice that the HEK293T clone KO1 expresses a residual low level of a

truncated form of Hop, which we characterized by mass spectrometry (MS) (Supplementary Table 1); it only retains the Hsp90-binding domain TPR2A." However, I do not see any discussion and consideration of this "low level of expression" in their results. Authors need to discuss this issue and show that this expression has no effect on their experiments.

3. Authors state that "We could see side views of single-capped 26S and double-capped 30S proteasome particles, and the two expected top views: an uncapped form with a central hole and a capped form corresponding to CP and RP face up, respectively (Fig. 4d and Supplementary Fig. 14 4g)." I found that it is very hard to see from raw TEM micrographs to observe this conclusion. Further "TEM image analysis" section in the methodology does not include any details (i.e. parameters used) to reproduce the analysis. This section requires attention: 1) all the 2D class averages should be presented rather than selected few of them and the raw data. 2) Details of the calculations should be given in the methodology; i.e. parameters, number of classes etc.

4. Homology modeling of Hsp90s with treading method does not work well and it is not convincing that the quality of the models is good. Also no validation has been done for residue level quality control. Nevertheless, for the purpose of this article, I guess what the authors are presenting here is acceptable, as they are just using these models to show the location of two residues that they regard as functionally important. However, their statement would be stronger if they link their statement (below) to existing structural analysis articles. Authors state that "As in bacteria, these residues are well surface-accessible in both human Hsp90 isoforms (Fig. 6f and Supplementary Fig. 7e), and modifying these residues in human Hsp90a strongly suppresses the interaction with Hsp70 in vitro (Fig. 6g) and with endogenous Hsp70 and Hsc70 in KO cells (Fig. 6h and Supplementary Fig. 7f)." Authors should look at the following article where functionally important Hsp90alpha residues are identified in different functional states by using perturbation response scanning and dynamic network analysis. The indicated two Hsp90alpha residues are also identified by Penkler et al. Although the current manuscript does not present strong bioinformatics aspects, linking their data to existing literature would improve their findings.

Penkler DL, Atilgan C, Tastan Bishop Ö. Allosteric Modulation of Human Hsp90α Conformational Dynamics. *J Chem Inf Model.* 2018;58(2):383-404. doi:10.1021/acs.jcim.7b00630

Reviewer #4 (Remarks to the Author):

I truly enjoyed reading this paper by Picard and collaborators. They use an elegant combination of techniques going all the way from bioinformatics to in vivo experiments, folding assays and interaction analysis to probe the roles of Hop/Sti1 on the refolding and degradation machineries in cells. The paper convincingly finds a link between the two mechanisms and rationalizes the effect of increasing the folding kinetics when proteasomal degradation is less efficient due to the deletion of Hop.

I am convinced this paper will have a big impact not only on the chaperone community but also on the wider area of folding and aggregation.

I suggest publication as it is.

Point-by-point response to the reviewers' comments

We appreciate the reviewers' thorough reading and highly constructive criticism. We have now performed a number of new experiments and restructured the paper, notably by disentangling and simplifying the main figures and moving more of the confirmatory evidence into the supplementary figures. Note that a new coauthor (Yashar Sadian) had to be added for the additional analysis of our EM data and that all bar graphs were redrawn to comply with the new Nature Communications rules. Other detailed responses are provided below. To facilitate the comparison with the originally submitted version of the manuscript, we provide a table at the end of this file as Annex 1, which maps the "old" to the "new" figures.

Reviewer #1 (Remarks to the Author):

In this study the authors attempt to characterize the role of the Hsp90 co-chaperone Hop by using a clever series of experiments/techniques including wide-scale proteome and interactome analysis as well as construction of several useful Indel CRISPR cell lines. From their analysis, they focus their attention on the proteasome system which is functionally impacted by loss of Hop. Despite a good premise, this work does not go far enough to characterize the molecular nature of the defect in proteasomal function. Many of the figures feel crowded with negative/control results that could be put in supplemental. Although the authors did a fantastic job with their proteomics analysis, they do not really explore the data in enough detail. Specific comments:

1) The foundation of this work rests on utilization of CRISPR KO cell lines. As the authors are aware, this technology does not produce clean knockouts as with micro-organisms. Several non-ideal situations can arise. First, a truncation of the protein can be produced (from insertion of a premature stop codon). Secondly, a frame shift mutation can occur which leaves a small part of the original protein and then after the frame shift a random amino acid sequence. Although the authors attempt address this by Western Blot (Fig. 1) and by analyzing peptides from WT and KO cells (Table S1), this is not a good enough validation. The genomic area around the gRNA binding site needs to be sequenced to fully characterize where the stop codon was introduced. Please note, this needs to be done for both copies of the genes involved. For more details please see <https://www.nature.com/articles/nprot.2013.143>

We agree that validating KOs is critical and cannot be taken lightly. While the indicated reference does not really provide any relevant information regarding this issue, we are convinced that mass spec is the best way to do it. After all, what one tries to achieve with a KO is to eliminate the protein. A more detailed genomic analysis would not be able to exclude that some protein is still being made through alternate promoter usage or alternative splicing. In contrast to that, we probe for the presence of recognizable protein in all KO lines by Western blot, and for peptides by mass spec

in 3 KO lines (two 293T KO lines and one HCT116 KO line). The second best way of checking is to look at the production of mRNA from the locus. We now include a QPCR experiment in Supplementary Fig. 1a, which fully supports for all KO lines what we see at the level of protein.

2)Figure 1: The authors spend a large amount of figure real estate on showing little difference between WT and KO cells (Fig. 1a-h). While important, the truly interesting data is the omic data in 1i. While it is clear that the majority of the proteome remains unchanged, there are proteins significantly altered in abundance. I recommend moving some of the earlier "control" figures 1a-h to supplemental, then making 1i larger with GO analysis on the most changed proteins. Correspondingly, there should be more analysis of 1i in the text.

As suggested, we moved a lot of the results into the supplementary figures. Incidentally, the fact that not much changes when Hop is absent is one of the key messages of our paper.

We incorporated a GO term enrichment analysis in revised Figure 1c. Please note that, while we see enrichment of some GO terms in KO cells, we don't see any statistically significant ($p < 0.05$) enrichment of biological processes in any of the WT cells used in this experiment.

3)Figure 3: The analysis of the Hop interactome is very exciting, but could be represented in a clearer manner:

-3a is really a control experiment and should be in supplemental. Instead, I recommend a schematic of the proteomics workflow for fig. 3a.

For this figure as well, some of the panels were moved to the supplementary figures. To avoid cluttering the main figure again, we refrained from including a proteomic workflow. After all, the experiment is straightforward and not much of a workflow.

-The pie chart in 1b is not clearly worded, please change to "1.57%, 10 enriched in K8A, K298A Hop".

We changed it.

-the notation for the double mutant seems odd. I would revise to either "AA Hop" or "K8A, K298A Hop".

We changed it to K8A,K229A Hop. Note that the mutation is K229A, not K298A.

-3c-the y-axis legend seems to have an extra "w" in it.

Corrected.

-3c This seems like an ineffective way to represent such nice data. At the very least I would change the y-axis to a log scale or even better, draw the network of interactions

and color them based on enrichment.

We changed the axis to log₂ scale.

-Given that the majority of the interactions go away in the mutant Hop that cannot bind Hsp90, are the authors detecting simply Hsp90 interactors rather than direct interactors of Hop? Rephrased another way, are you detecting Hsp90-client complexes present when Hop is bound to Hsp90? There needs to be some clarity in the text on this.

We clarified the statement in the text to indicate that Hop "interactors" largely appear to be interactors of Hsp70 and/or Hsp90 rather than direct interactors.

-Some of the important interactions detected in the MS experiment should be validated in vivo (with IP-Western Blots) from both HEK cells and in vitro (IP-Western with recombinant versions of the proteins) to ascertain direct vs bridged interactions.

As mentioned in our response to the previous comment, Hop interactions are largely indirect. Exploring this by in vitro experiments would be difficult to do from a technical point of view (and beyond the scope of this study) and difficult to justify at this point. Moreover, we would like to point out that we also show proteomic data of proteins associated with the purified proteasome. This provides an independent validation. However, as suggested, we now include standard co-IP data showing that several proteasome components are indeed associated with Hop-containing complexes, directly or indirectly (new Figure 3d).

4) Figure S3K. The authors mentions KO of Hsp90a and b. Given that there is disagreement in the literature on the essential nature of Hsp90 isoforms, the authors need to sequence their KO cells as in comment 1.

These results are merely complementary. The detailed characterization of these KOs will be the subject of a different publication. Having said that, we have of course both Western and proteomic data to support the conclusion that the KOs are KOs. We incorporated Western blot data in support of the Hsp90 KO lines in revised Supplementary Figure 4f. Please note that there is absolutely no disagreement on the essential nature of Hsp90 isoforms in the literature; we would like to refer the reviewer to the Hsp90 facts sheet on the Picard lab web page.

5) Figure 4: As with many of the figures, Figure 4 seems unnecessarily crowded with control/negative results. An example is 4c, which is not showing any impact of the knockout on proteasomal components. Readers can simply refer to the data in a supplementary table. The key take-home message of this figure is that proteome capping may be altered in Figure 4e, but is not convincing.

We have restructured Figure 4. However, again, one of the key points of the paper is that most things don't change because cells maintain proteostasis otherwise. Regarding Figure 4e (now Figure 5a-d), in response to comment 3 of reviewer 3, we provide more details on the quantitation of the TEM data in the revised manuscript.

6) Overall, we suggest that the authors look at possibly re-arranging figures 3 and 4. Figure 3 should really be a clear analysis of the Hop interactome (Hsp90-dependent and independent) and Figure 4 should focus on the proteasome. At the moment, the foci of these figures is blurred and overlapping.

Figures have now been rearranged as suggested. However, since there may not be any genuine Hsp90-independent Hop interactors (see above), this is not further followed up.

7) The authors try to examine the exact nature of the proteasomal defect. They clearly show steady state levels of the proteasomal subunits are not altered and show unconvincing data on the formation of the proteasome complex. There are other potential ways the proteasome could be impacted. For example, are PTMs on the individual proteasomal subunits altered in cells lacking Hop? The protein levels may be unchanged but PTMs may be different. This can easily be examined by the kind of proteomics the authors demonstrate in the paper.

We actually discuss the possible involvement of PTMs in the second paragraph of the Discussion. As to identifying PTMs and their changes "easily...by the kind of proteomics", we are a bit at a loss to understand how this could be an easy task. Typically, directed proteomics is required for this and even then, it cannot easily uncover all imaginable PTMs. In Annex 2 at the end of this file, we provide a more in-depth analysis of this issue and what's known about PTMs for the reviewer.

8) Alternatively, the subunits may assemble fine in vitro but not in vivo. Is it possible that the localization of the proteasomal subunits is altered in Hop KO cells?

Regarding the first comment, we do not understand which subunits are meant and why one would suspect a difference between in vitro and in vivo and what "alternatively" refers to. Whether the localization of proteasomal subunits is altered is an interesting question, which we have now explored and included in Supplementary Fig. 6i. The result is that they aren't.

Reviewer #2 (Remarks to the Author):

In this paper, the authors address the role of Hop, a Hsp90 and Hsp70 cochaperone, in proteostasis. They present data showing that Hop knockout cells are unable to efficiently cap the core particles of the proteasome with regulatory particles and thus are inefficient at degradation compared to WT cells. They further show that knockout cells are better able to prevent aggregation and promote protein folding than WT cells. The results suggest that the one role of Hop is to shift the proteostatic balance from refolding towards degradation.

A great deal of work has gone into this project and the conclusions are supported by the data presented. The results and conclusions are exciting and will be of interest to specialists in the field of chaperones as well as to a general audience. However, one

major comment is that it is not written for a general audience, but rather for specialists in the field. The impact of the most important conclusions is diminished by being mingled with the other observations, negative results and less important conclusions and suggestions.

As mentioned above, we have followed the advice to simplify the figures and to move more material into the supplementary figures. The text has been adapted to match these major changes. We do hope that the reviewer will find the paper more readable, and even though it is of course a research paper, intelligible for the *Nature Communications* readership.

Comments:

1. Overall the paper could be significantly improved by writing in a clearer, more concise manner with the emphasis on the major findings. For example, the first section could go in the supplementary information or be moved to a less prominent place in the manuscript, since much of the data is negative and might have been expected from previous yeast results. The conclusion could be stated with a reference to the SI.

Hopefully things will be clearer with the restructured manuscript. This is notably true for Fig. 1. Many of its descriptive results have been moved to Supplementary Fig. 1. However, we do not agree that we could have expected all or most of these results from what was known from yeast. For example, that Hop KO could be more resistant to heat-shock could not be predicted from yeast; at the time, what had been reported was that *Δsti1* yeast are cold- and heat-sensitive. Of course, we hope that we have given due credit to the more recent published work of the reviewer's lab (which supports and complements our conclusions).

2. Overall the figures could be clearer. Overall, many of the panels would be easier to follow if they were better labeled. Without looking up instructions to authors, the figures appear to have too many panels per figure. Figs. 1, 2, 3, 4, 5 and 6 have so many panels that it is difficult (impossible) to read the words when printed on paper.

Corrected.

3. In general, figure legends could be improved by adding necessary information, so the reader understands what is shown. The reader has to go back to the methods to find out what components are in the experiments and how many replicates were performed.

We apologize for this telegraphic style, but perhaps the reviewer is unaware of the fact that there is a word limit for the legends. Regarding the replicates, now we show all data points in the revised figures.

4. For many experiments, data with two knockout strains are presented. It is important to show that there is not just one unique knockout. However, it might be easier for the reader if some of the data for the two strains were put in the SI and if the figures had fewer panels. The same applies for the yeast strains.

For the revised manuscript we moved most of this into the supplementary figures. For yeast it should nevertheless be pointed out that there are potentially relevant differences between the two strain backgrounds. In our hands, only W303 shows the ts phenotype expected from the original report from Betty Craig's lab, and yet, luciferase recovers faster in it as well.

5. Perhaps the wording of the last sentence of the abstract could be improved. "Thus, cells may act on Hop to shift the proteostatic balance between folding and degradation." It is not that cells are acting on Hop, but rather regulatory proteins.

Thanks for pointing out that something was not quite right. We have revised the sentence.

6. The meaning of "axis" in the section heading for section 2 is not clear.

We deleted the term from the revised text.

7. Fig. 1a. Move Hop to line up with band.

To avoid confusion, we have now added a bracket.

8. The cartoon in Fig. 2A is confusing. How can aggregates of "misfolded proteins" get converted to "disaggregated peptides". One choice is to change "peptide" to "proteins", since peptides are less than 50 amino acids. However, how are "disaggregated proteins" different than "misfolded proteins". The cartoon shows that they are. It might be clearer if the arrow from aggregates went back to the pool of "misfolded proteins" and that pool was referred to as "unfolded and misfolded proteins." Since it is called "a proteostatic equilibrium" in the legend, perhaps some arrows should go in both directions.

Thank you for pointing that out. The revised figure shows a revised cartoon.

9. Fig. 3b. Label as "Hop interactors". Check significance of numbers (1.57%). Eliminate the number for each category, since the total number is given.

While we prefer to avoid the term "Hop interactors" since most/all are indirect ones (see also our response to comment 3 of reviewer 1), we have eliminated the numbers as suggested.

10. Fig. 3d. The pie graph is not necessary.

Deleted.

11. Fig. 3e, f and h. Not every KO strain needs to be shown and the rest could go in SI.

The figure was restructured.

12. Fig. 4a, change title to something like "fold change (log2) following incubation with whatever Hsp90 inhibitor was used" (it is not mentioned in the text or the fig. legend).

Changed in figure and corresponding legend.

13. Fig. 4b, c, d could go in SI.

The figure was restructured.

14. Fig. 4e. It is confusing what the difference is between capped and 30S particles? Are the two graphs showing different things?

In response to a specific comment (#3) of reviewer 3, we have completely redone the corresponding figure (images) and present the quantitation differently (hopefully more clearly). This is now in Fig. 5.

15. Fig. 4g. One strain could be moved to the SI.

The figure was restructured.

16. Fig. 5a, k and l could be moved to the SI.

The figure was restructured.

17. Fig. 6a. Cartoon is not clear. Perhaps show the pathway consistent with the results and leave out the "expectations" since they were not consistent with the data. It would be helpful to label the cartoon for clarity (90, 70, Hop, J).

The cartoon is meant to help the reader follow the story. We did improve the readability by adding labels as suggested.

18. Fig. 7, g and h cannot be compared since different concentrations of proteins were used and the data are plotted differently.
Sue Wickner

The two panels address different questions. They do not need to be compared one on one.

Reviewer #3 (Remarks to the Author):

I was asked to review the structural and bioinformatics aspects of this manuscript. I read through the whole article though, to form a general opinion. In general, I found that the manuscript is very clearly and logically written. The scientific questions are well put and the experiments are well designed.

In my opinion the article is acceptable subject to the following corrections:

1. In Introduction section, authors state that "Hop forms a ternary complex with Hsp70 and Hsp90 using its tetratricopeptide repeat (TPR) domains. Two of its three TPRs, TPR1 and TPR2A, specifically bind the extreme C-terminal sequences EEVD and MEEVD of Hsp70 and Hsp90, respectively." However, the literature reports other potential interactions between Hop and Hsps; including Hop-Hsp90 middle domain interactions. Hence, authors should cover all the potential interactions between these proteins as reported previously. Thus, their references need to include at least these two articles:

A.B. Schmid, S. Lagleder, M.A. Gräwert, A. Röhl, F. Hagn, S.K. Wandinger, et al., The architecture of functional modules in the Hsp90 co-chaperone Sti1/Hop, *EMBO J.* 31 (2012) 1506–1517.

Hatherley R, Clitheroe CL, Faya N, Tastan Bishop Ö. Plasmodium falciparum Hop: detailed analysis on complex formation with Hsp70 and Hsp90. *Biochem Biophys Res Commun.* 2015;456(1):440-445.

Indeed, it is appropriate to mention these additional contacts, and we are happy to take advantage of this revision to correct this.

2. Authors indicate that "We did notice that the HEK293T clone KO1 expresses a residual low level of a truncated form of Hop, which we characterized by mass spectrometry (MS) (Supplementary Table 1); it only retains the Hsp90-binding domain TPR2A." However, I do not see any discussion and consideration of this "low level of expression" in their results. Authors need to discuss this issue and show that this expression has no effect on their experiments.

We actually did discuss that, in the first paragraph of the Results and then again in the third paragraph of the Discussion.

3. Authors state that "We could see side views of single-capped 26S and double-capped 30S proteasome particles, and the two expected top views: an uncapped form with a central hole and a capped form corresponding to CP and RP face up, respectively (Fig. 4d and Supplementary Fig. 14 4g)." I found that it is very hard to see from raw TEM micrographs to observe this conclusion. Further "TEM image analysis" section in the methodology does not include any details (i.e. parameters used) to reproduce the analysis. This section requires attention: 1) all the 2D class averages should be presented rather than selected few of them and the raw data. 2) Details of the calculations should be given in the methodology; i.e. parameters, number of classes etc.

Good point. We have reanalyzed our samples to include a much larger number of particles and to generate statistically stronger conclusions. These are now shown in the new Fig. 5a-d. We also provide more details on the analysis in the Methods. "Supplementary Fig. S14 4g", which presumably meant Supplementary Fig. 4g, has been removed from the revised manuscript to avoid overloading the manuscript with too many duplicate experiments in other cell lines (in keeping with a comment of

reviewer #2).

4. Homology modeling of Hsp90s with treading method does not work well and it is not convincing that the quality of the models is good. Also no validation has been done for residue level quality control. Nevertheless, for the purpose of this article, I guess what the authors are presenting here is acceptable, as they are just using these models to show the location of two residues that they regard as functionally important. However, their statement would be stronger if they link their statement (below) to existing structural analysis articles. Authors state that "As in bacteria, these residues are well surface-accessible in both human Hsp90 isoforms (Fig. 6f and Supplementary Fig. 7e), and modifying these residues in human Hsp90a strongly suppresses the interaction with Hsp70 in vitro (Fig. 6g) and with endogenous Hsp70 and Hsc70 in KO cells (Fig. 6h and Supplementary Fig. 7f)." Authors should look at the following article where functionally important Hsp90alpha residues are identified in different functional states by using perturbation response scanning and dynamic network analysis. The indicated two Hsp90alpha residues are also identified by Penkler et al. Although the current manuscript does not present strong bioinformatics aspects, linking their data to existing literature would improve their findings.

Penkler DL, Atilgan C, Tastan Bishop Ö. Allosteric Modulation of Human Hsp90a Conformational Dynamics. *J Chem Inf Model.* 2018;58(2):383-404.
doi:10.1021/acs.jcim.7b00630

We appreciate the reviewer's conclusion that our modelling is acceptable. We are happy to mention the supporting evidence published by Penkler et al. in the relevant section of the revised Methods.

Reviewer #4 (Remarks to the Author):

I truly enjoyed reading this paper by Picard and collaborators. They use an elegant combination of techniques going all the way from bioinformatics to in vivo experiments, folding assays and interaction analysis to probe the roles of Hop/Sti1 on the refolding and degradation machineries in cells.

The paper convincingly finds a link between the two mechanisms and rationalizes the effect of increasing the folding kinetics when proteasomal degradation is less efficient due to the deletion of Hop.

I am convinced this paper will have a big impact not only on the chaperone community but also on the wider area of folding and aggregation.

I suggest publication as it is.

Annex 1: Correspondence between "old" and "new" figures

Figure/panel no. in first submission	Figure/panel no. in the revised manuscript
1b	S1b
1c	S1c
1d+S1a+S1b	S1d
1e+S1c+S1d	S1e
1f	1d
1g (HEK293T panel)	S1g
1g (HCT and A549 panels)	1e
1h	1f
1i	1b
	1c (new)
S1e	S1f
S1f	S1h
	S1a (new)
S2c	S2b
S2d	S2c
S2e	S2d
S2f	S2e
2d	2d+S2f
2e	2e+S2h
S2h	S2f
S2k	S2i
3a	S3a
S3a	S3b
3b	3a
3c	3b
3d	3c
	3d (new)
S3b	S3c
S3c	S3d
S3d	S3e
S3e	S3f
3e	4a+S4a
3f	4b+S4b
3g	4c
3h	4d
3i	S4g
3j	4e
S3f	S4a
S3g	S4b
S3h	S4c

S3i	S4d
S3j	S4e
S3k	S4f
S3l	S4h
S3m	S4i
4a	4f
4b	4g
4c	4h
	5a (new)
	5b (new)
	5c (new)
	5d (new)
4f	5e
S4a	S5a
S4b	S5b
S4c	S5c
S4d	S5d
S4e	S5e
S4f	S5f
4g	5f+S6a
4h	5g+S6f
4i	5h
S4i	S6b
S4j	S6c
S4k	S6d
S4l	S6e
S4m	S6g
	S6h (new)
	S6i (new)
4j	S7c
4k	S7g
4l	S7h
S5a	S7a
S5b	S7b
S5c	S7d
S5d	S7e
S5e	S7f
5a	S8a
5b	6a
5c	6b
5d	6c
5e	6d
5f	6e
5g	6f
5h	6g
S6a	S8b

S6b	S8c
S6c	S8d
S6d	S8e
S6e	S8f
S6f	S8g
S6g	S8h
5i	7a
5j	7b
5k	S9b
5l	S9d
5m	7e
5n	7f
S6h	S9a
S6i	S9c
S6j	7c
S6k	S9e
S6l	7d
S6m	S9f
S6n	S9g
S6o	S9h
6a	8a
6b	8b
6c	8c
6d	8d
6e	8e
6f	8f
6g	8g
6h	8h
S7a	S10a
S7b	S10b
S7c	S10c
S7d	S10d
S7e	S10e
S7f	S10f
S7g	S10g
S7h	S10h
7a	9a
7b	9b
7c	9c
7d	9d

Annex 2: PTMs in proteasome and their identification by MS

A wide range of PTMs have been reported for proteasome subunits, including acetylation, ADP-ribosylation, glutathionylation, phosphorylation, ubiquitination, S-nitrosylation (reviewed by Kors et al , Front Mol Biosci., 2019 Jul 16;6:48). For phosphorylation alone, more than 400 modification events are listed in databases such as Phosphosite.org (reviewed by Guo X, et al. Protein Cell. 2017, Apr;8(4): 255-272). Despite this wealth of identification data, for the overwhelming majority of these PTM's there is simply no information available on their impact on proteasome activity.

In practice, the study of many of these PTMs requires dedicated sample preparation protocols using specific inhibitors to preserve labile groups or enrichment steps. A thorough characterization of all proteasome PTMs in WT and Hop KO cells would be a complex, long term project in itself, going way beyond the scope of this study. Furthermore, as the protocol available only purifies the pool of capped 26S proteasome (which is presumably functional in both genotypes), we would not expect to find striking differences in modifications in our samples.

We have nevertheless reprocessed the available MS data on whole proteasome fractions from WT and Hop KO cells, considering as possible PTMs phosphorylation, Lys acetylation, Ubiquitination and myristoylation (reported for one specific subunit, PSMC1=PRS4). Each type of PTM was identified and quantitated with MaxQuant.

As is often the case, the identification of PTM's in non-enriched fractions yields sparse results (see table). No ubiquitination sites were identified, which is not surprising as such peptides are difficult to identify by MS without enrichment, even in medium complexity samples. Only a few phospho-sites and acetylation sites were found (Table) but not all of them were reproducibly detected in both cell lines.

PTM	Protein, site	HEK (log ₂ (fold change))	HCT (log ₂ (fold change))	Reported effect of PTM on activity
Phospho	PSMC3 - Ser9	-1.27	-0.08	Unknown
Phospho	PSMC4 - Thr 311	0.52	0.29	Unknown
Phospho	PSMC4 - Thr 316	-0.35	0.29	Unknown
Phospho	PSMD1 -Thr 311	0.04	0.11	Unknown
Phospho	PSMD1 - Thr273	0.09	-0.22	Inhibition
Phospho	PSMD1 - Ser 315 (2xP)	-0.26	nd	Unknown
Phospho	PSMD1 -Thr 311 (2xP)	-0.26	nd	Unknown
Phospho	PSMD11 - Ser 14	-0.66	nd	Activation
Phospho	PSMD2 - Ser 16	-0.69	-0.21	Unknown
Phospho	PSMD2 - Ser 361	-0.85	-0.43	Unknown
Phospho	PSMD4 - Ser 266	-0.18	-0.20	Unknown
Acetyl (K)	PSMC4 - Lys 409	-0.04	nd	Unknown
Acetyl (K)	PSMD1 -Lys 868	0.20	nd	Unknown
Acetyl (K)	PSMD1 - Lys 869	0.20	nd	Unknown

Acetyl (K)	PSMD2- Lys 41	0.42	nd	Unknown
Acetyl (K)	PSMD1 -Lys 310	nd	-0.38	Unknown
Acetyl (K)	PSMD3-Lys16	nd	-0.25	Unknown
Myristoyl (G)	PSMC1-Gly 2	-0.29	0.13	Nuclear localization?

After normalization by total protein amounts, the fold changes between WT and KO are generally small or very small, suggesting only minor changes. Overall, the data does not suggest that there would be strong changes in PTMs in the 26S particles that are detectable easily without in-depth, dedicated investigations.

REVIEWERS' COMMENTS

Reviewer #1 (Remarks to the Author):

The authors have made a concerted effort to address all the reviewer comments and in doing so have produced a substantially improved manuscript. We congratulate the authors on their elegant study and look forward to its publication in Nature Communications.

Andrew Truman

Reviewer #2 (Remarks to the Author):

The revised manuscript is much improved and the reviewers' suggestions and comments have been satisfactorily addressed.